# Complex transcriptional regulations of a hyperparasitic quadripartite system in giant viruses infecting protists

Alexandra Bessenay ®[1], Hugo Bisio ®[1], Lucid Belmudes[2], Yohann Couté ®[2], Lionel Bertaux ®[1,3], Jean-Michel Claverie ®[1], Chantal Abergel[1], Sandra Jeudy ®[1] ✉ & Matthieu Legendre ®[1] ✉

Hyperparasitism is a common pattern in nature that is not limited to cellular organisms. Giant viruses infecting protists can be hyperparasitized by smaller ones named virophages. In addition, both may carry episomal DNA molecules known as transpovirons in their particles. They all share transcriptional regulatory elements that dictate the expression of their genes within viral factories built by giant viruses in the host cytoplasm. This suggests the existence of interactions between their respective transcriptional networks. Here we investigated *Acanthamoeba castellanii* cells infected by a giant virus (megavirus chilensis), and coinfected with a virophage (zamilon vitis) and/or a transpoviron (megavirus vitis transpoviron). Infectious cycles were monitored through time-course RNA sequencing to decipher the transcriptional program of each partner and its impact on the gene expression of the others. We found highly diverse transcriptional responses. While the giant virus drastically reshaped the host cell transcriptome, the transpoviron had no effect on the gene expression of any of the players. In contrast, the virophage strongly modified the giant virus gene expression, albeit transiently, without altering the protein composition of mature viral particles. The virophage also induced the overexpression of transpoviron genes, likely through the indirect upregulation of giant virus-encoded transcription factors. Together, these analyses document the intricated transcriptionally regulated networks taking place in the infected cell.

Hyperparasitism is a phenomenon whereby one parasite is itself targeted by another, leading to a complex network of interactions within the host. A variety of pathogenic species can be hyperparasitized, including eucaryotes[1] and prokaryotes[2]. These tripartite host-parasite-hyperparasite interactions may lead to different outcomes during their coevolution, such as hypovirulence when the hyperparasite attenuates the virulence of the parasite, or, conversely, hypervirulence[3,4]. From an ecological perspective, hyperparasitism regulates host-parasite cycles and, therefore, influences host population dynamics[3–6].

[1]Aix-Marseille University, Centre National de la Recherche Scientifique, Information Génomique & Structurale (IGS), Unité Mixte de Recherche 7256 (Institut de Microbiologie de la Méditerranée, FR3479), IM2B, IOM, Marseille, Cedex 9, France. [2]Univ. Grenoble Alpes, INSERM, CEA, UA13 BGE, CNRS, CEA, FR2048, Grenoble, France. [3]Present address: Aix-Marseille University, Centre National de la Recherche Scientifique, Laboratoire de Chimie Bactérienne (LCB), Unité Mixte de Recherche 7283 (Institut de Microbiologie de la Méditerranée, FR3479), IM2B, Marseille, France. ✉e-mail: jeudy@igs.cnrs-mrs.fr; legendre@igs.cnrs-mrs.fr

Although viruses are often considered the ultimate parasites, some so-called giant viruses are also the target of their own parasites. Giant viruses infect a wide range of unicellular eukaryotes[7] and have large dsDNA genomes of up to 2.5 Mb[8] encapsidated in viral particles of up to 1.5 μm in length[9]. Some of them are infected by other viruses called virophages, that exhibit much smaller icosahedral capsids of 50 to 75 nm in diameter, and dsDNA genomes of 17 to 30 kb[10–14]. After entering the host cell independently of the giant virus[15] or concomitantly[10,12], virophages hijack the viral factory (VF), a transient organelle initiated by the giant virus where genome replication and viral particles production take place. Interestingly, virophages may either inhibit the production of giant virus particles[10,11] or have no apparent deleterious effect[12,16]. Virophages can also integrate into the host cellular genome in a provirophage (transcriptionally silent) state and then protect the host cell population upon subsequent reactivation[15]. Numerous intertwined interactions are therefore observed in hyperparasitic systems involving cells, giant viruses and virophages. These have a potentially significant ecological impact, as a wide diversity of giant viruses[17–20] and virophages[5,21,22] has been uncovered by metagenomics in various environments.

In addition to the tripartite hyperparasite system, an extra layer of complexity is also observed within the *Megamimivirinae* subfamily of giant viruses. Indeed, some of them are found associated with plasmid-like linear dsDNA molecules of 7 kb. Such molecules coding for 6 to 8 genes are named transpovirons. These can be found as episomes both within the giant viruses and virophages particles, but can also integrate their genomes[23]. They are suspected of replicating in the VF of giant viruses and spreading using both giant viruses and virophages as carriers[16].

Importantly, the expression of both virophages and transpovirons genes is dependent on the transcription machinery encoded by the giant virus[16,24], and is therefore controlled by its specific transcriptional regulatory elements[11,25,26] (Supplementary Data 1). This suggests tight, yet undetermined, interactions at the transcriptional level between giant viruses, virophages and transpovirons in the context of host cell infection.

In this work, we sought to decipher the transcriptional program governing this entangled hyperparasitic system and determine the effect of each partner on the others. We thus performed an RNA-seq time-course experiment of *Acanthamoeba castellanii* cells (C) infected by the giant virus (GV) megavirus chilensis[27], then added the other players: the zamilon vitis virophage (Vp)[16] and the megavirus vitis transpoviron (Tpv)[16]. Our experiments revealed the transcriptional program of each player during the infectious cycle, as well as a wide range of transcriptional responses to the various interactions.

## Results

### *A. castellanii* host (co-)infections by megavirus chilensis giant virus, zamilon vitis virophage and megavirus vitis transpoviron

To study the transcriptional impact of the different players of this hyperparasitic system, we used 4 different infection setups (Fig. 1A). *A. castellanii* cells were infected with megavirus chilensis, a GV devoid of an associated Vp or Tpv[16]. The results were compared to 3 additional coinfection experiments, adding the Vp or the Tpv, or both (Fig. 1A). All 4 conditions were followed during a complete GV infectious cycle, with RNA samples collected at the same timepoints from 30 min to 12 h post infection (pi), resulting in 6 samples per condition (Fig. 1B). In addition, mock-infected cells were included as controls using heat-inactivated giant viruses and associated players (see "Methods"). All experiments were carried out in three biological replicates.

A total of 82 polyA-enriched RNA samples were successfully sequenced, resulting in an average of 15.8 million read pairs per sample, of which 98.5% passed quality control (Supplementary Data 2A). These were mapped on the reference genomes of *A. castellanii* (C), zamilon vitis (Vp), megavirus vitis transpoviron (Tpv) and megavirus chilensis (GV). Of note, we reassembled the genomic sequence of the latter (see "Methods"), which added 19,414 bp terminal inverted repeats (TIRs) to the previously reported sequence[27]. The genomic structure is similar to other viruses from the same genus[28] (Figure S1).

Most of the read pairs (mean = 85.4%, sd = 1.7%, Supplementary Data 2B) were successfully aligned to a combined reference gene set of

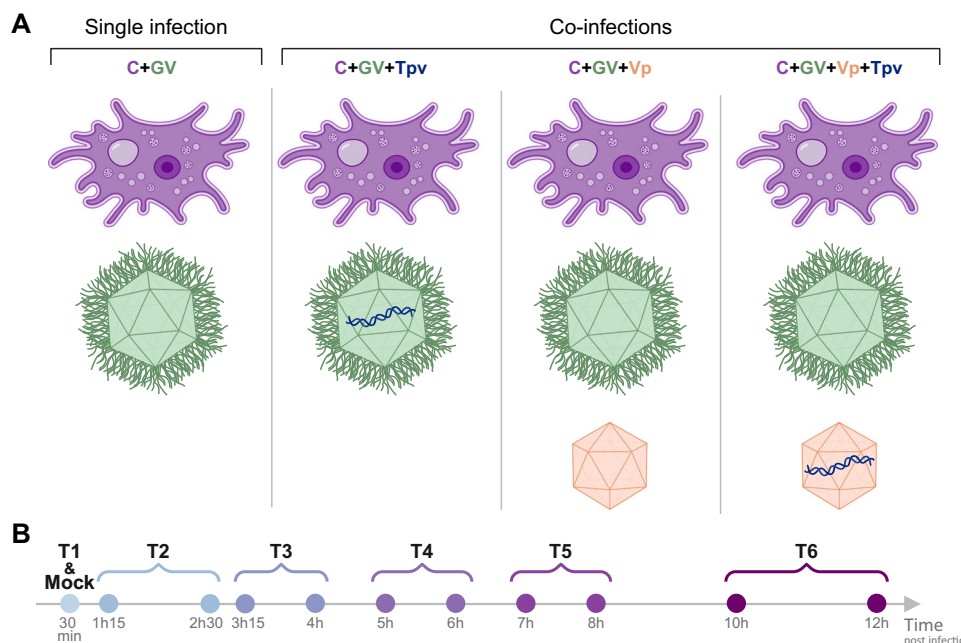

**Fig. 1 | Schematic diagram of (co-)infection experiments. A** Schematic representation of the 4 infection experiment conditions (in columns) for which RNA-seq was performed. Four partners are involved: *A. castellanii* (C: purple), megavirus chilensis (GV: green), zamilon vitis (Vp: orange) and megavirus vitis transpoviron (Tpv: blue). The players involved in each condition are indicated at the top of each column. **B** The timeline summarizes the time points at which RNA samples were collected, with the time post-infection indicated below each collection point. Prior RNA extraction, some samples were pooled, as indicated by braces, and will be subsequently referred to by the names shown at the top (from T1 to T6, and mock). Icons representing partners were created with BioRender.com.

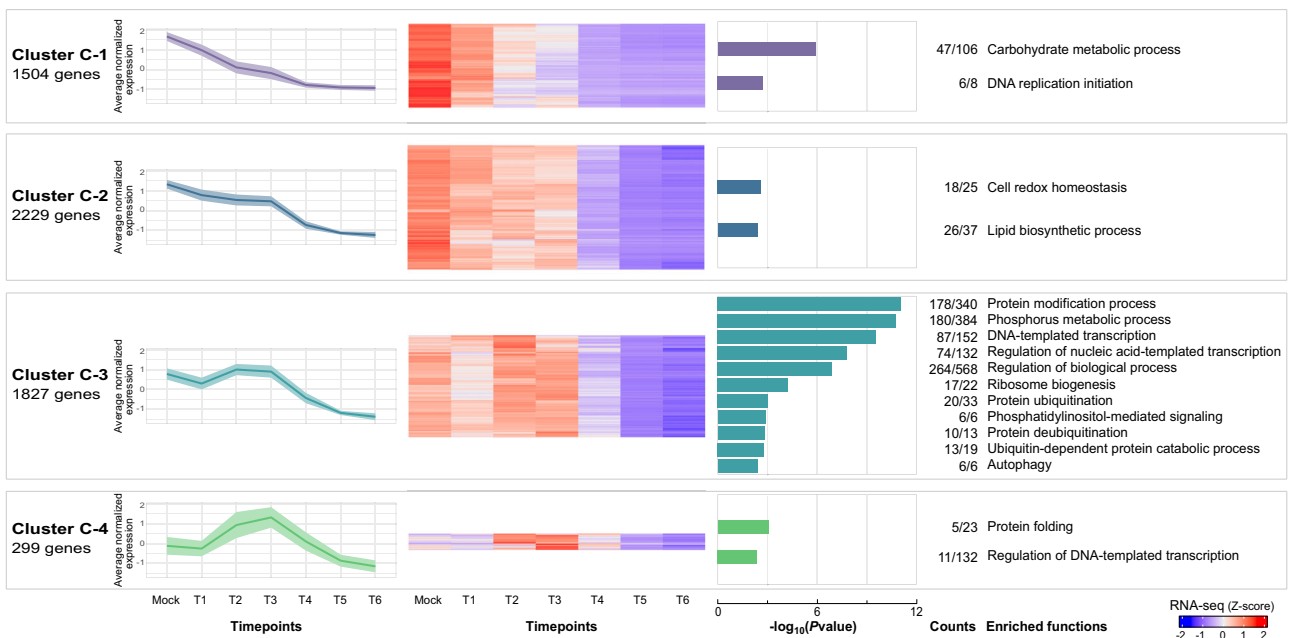

**Fig. 2 | Transcriptional patterns of differentially expressed *A. castellanii* genes during megavirus chilensis infectious cycle.** Hkmeans clustering of the 5859 cellular genes differentially expressed between mock and GV infection. The left-most part of the figure shows clusters names and the total number of genes associated to each cluster. Average normalized expression (Z-score of log₂-transformed TPM values) over time post-infection is then showed for each cluster (solid lines), as well as the corresponding standard deviation (colored areas). The heat-maps show the expression patterns (Z-scores of log₂-transformed TPM values) of each gene (rows) along the different timepoints (columns), averaged over the replicates. Colored histograms display one-sided Fisher Exact test *P*values of the biological processes (GO terms) significantly enriched in each cluster (*P*values ≤ 0.005). The numbers on the right indicate the number of genes with a given GO term (in cluster vs all differentially expressed genes). Functional annotations are displayed on the rightmost part of the figure. Source data are provided as a Source Data file.

all partners present. In each condition, all present players were detected, with a maximum proportion of mapped reads of 100%, 98.3%, 18.3% and 1%, for C, GV, Vp and Tpv, respectively (Figure S2). After filtration for low expression (see "Methods"), we also found that the vast majority of the annotated genes were expressed, with 82.7% for C (12845/15532), 98.2% for GV (1134/1155), and 100% for Vp and Tpv (with 20/20 and 7/7, respectively). Additionally, saturation curves show that sequencing depth was adequate to capture the gene expression dynamics of each individual partner (Figure S3).

## The *A. castellanii* host transcriptome is reprogrammed by megavirus chilensis infection

We first explored the transcriptional response of the *A. castellanii* host to its infection with GV alone. As shown by principal component analysis (PCA) of normalized host genes expression values (in transcript per million, TPM), the biological replicates are robust as they cluster together at each timepoint (Figure S4A). Moreover, infection time is the main source of variance since timepoint-specific segregation of the samples was observed. Importantly, mock and T1 timepoints do not overlap, demonstrating that the host transcriptome is disrupted as early as 30 min pi. By contrast, the proximity of T2 and T3 timepoints, as well as T5 and T6, indicates that there is no global shift in host transcription between 1h15min and 4 h pi, and from 7 h pi to the end of the infectious cycle.

We first analyzed the host transcriptome at the beginning of the infectious cycle by comparing mock and productive GV infection (T1). This revealed 347 differentially expressed genes (FDR *P*values < 0.01 and |log₂(Fold Change)| ≥ 1.5). Among them, the 162 whose expression increases at T1 are mainly involved in signal transduction (*P*value = $1.3 \times 10^{-4}$, Supplementary Data 3A) and are enriched in Rho family small GTPases (Supplementary Data 4A). These proteins are potentially linked to cytoskeletal remodeling at the initial stages of infection[29]. We also found 6 kinesins among the 185 genes significantly

underexpressed at T1, supporting the microtubule-based movement gene ontology (GO) term enrichment (*P*value = $6.1 \times 10^{-6}$, Supplementary Data 4A). Finally, nucleosome assembly is also potentially disrupted by infection (GO term *P*value = $1.5 \times 10^{-3}$), as 3 cellular histones (1 H1 and 2 H4, Supplementary Data 4A) are also underexpressed at T1 compared to mock infection.

We next expanded the analysis by comparing all timepoints (T1 to T6) to mock infection and found that from 2.7% (T1 vs mock) to 26.8% (T4 vs mock) of the host's expressed genes were differentially expressed. Combined together, we identified a total of 5859 differentially expressed genes between at least one timepoint and mock, which corresponds to 45.6% of the expressed host genes. Hybrid hierarchical k-means (hkmeans) clustering of those genes (k = 4) shows specific expression patterns and functions (Fig. 2 and Supplementary Data 3B). In addition, we scrutinized the *A. castellanii* promoter sequences to identify enriched motifs around transcription start sites (TSS) in relation to expression timing (Figure S5).

The first cluster (C-1), with gradually decreased expression over the time-course, is enriched in carbohydrate metabolism, specifically galactose metabolism (Fig. 2 and Supplementary Data 4B). Galactose is a major component of the *A. castellanii* cyst wall[30]. Here we found that genes associated with encystment, such as CSP21 (BAESF_04785) and encystation-mediating serine proteinase (BAESF_02870), are either weakly expressed or exhibit reduced transcriptional levels over time (Supplementary Data 5A). Thus, like mimivirus, megavirus likely represses encystment-mediating genes that would prevent viral infection[31]. C-1 cluster also contains the majority of licensing factors involved in DNA replication initiation (5/6) and a cell division control protein (CDC45-like), suggesting an arrest of the cell cycle upon infection. Promoters of genes found in this cluster are enriched in 4 motifs that resemble known transcription factors (TF) binding sites (Figure S5). One corresponds to E2F transcription factor and two others to HAP2 that recognizes CCAAT-box motifs.

The second cluster (C-2) shows a similar profile but with stabilized expression from T1 to T3. Thus, viral infection induces steady expression of cellular genes that are mostly involved in cell redox homeostasis (including numerous glutathione and thioredoxin reductases) and in lipid metabolism (Supplementary Data 4B).

In the third cluster (C-3), genes are first slightly repressed by the infection, then upregulated in T2 and T3 (1h15min to 4 h pi) to recover their basal expression level, before steadily decreasing again. They span various cellular functions that include 6 genes involved in autophagy (Supplementary Data 4B), a cellular process frequently deployed to restrict viral infections[32]. Numerous genes involved in protein modifications such as phosphorylation/dephosphorylation and ubiquitination/deubiquitination are also activated (Fig. 2). The manipulation of the host ubiquitin system to promote viral replication is widespread in viruses, in particular giant viruses from the *Nucleocytoviricota* phylum[33,34]. Although giant viruses strikingly encode translation-related genes[35], they remain dependent on the cellular host ribosomes for protein synthesis. Accordingly, we found cellular translation-related genes, several of which are involved in ribosome biogenesis and maturation, also enriched in this cluster. Surprisingly, we also found 87 transcription-related genes enriched in this cluster, including several units of the RNA polymerase I, II, and III, as well as transcription factors (RFX).

Finally, the last cluster (C-4), showing strong activation at T2 and T3 (from 1h15min to 4 h pi) also contains transcription factors in addition to chaperone proteins (DnaJ and HSP90). The latter may be part of the cellular stress response induced by the infection, or are specifically activated to support viral proteins folding[36]. In this cluster, gene promoters are enriched in two motifs, one of which corresponds to STAT transcription factor binding sites (Figure S5).

Taken together, these patterns show that the *A. castellanii* transcriptome is strongly reshaped by megavirus infection. Although all cellular genes are relatively underexpressed by the end of the infectious cycle, numerous functions, counting for a third of the expressed genes (4355/12845), are either maintained (cluster C-2) or activated (clusters C-3 and C-4) at various degrees. Several forces are probably at play here. The host likely responses to viral infection by activating general stress factors and more specific immune mechanisms[37]. But we also observed specific functions triggered to support viral replication, in line with the concept of the cell transforming into virocell[38,39].

A similar transcriptional reprogramming of the host has been observed in *Acanthameoba polyphaga*, a related amoeba from the same genus, infected by mimivirus[40]. Specifically, *A. polyphaga* genes involved in DNA replication and cytoskeletal remodeling are underexpressed during the mimivirus replication cycle. Similarly, *Acanthameoba* genes involved in transcription, translation regulation, and proteasome are activated in both *A. castellanii*/megavirus and *A. polyphaga*/mimivirus infections. However, several cellular genes associated with ribosome maturation, autophagy and protein folding are exclusively activated in the present study. This is likely due to an increased sequencing depth and temporal resolution of our host transcriptome analysis.

## Temporal dynamics of megavirus chilensis gene expression

We next analyzed the expression dynamics of the GV genes. As expected, almost all ($n = 1098$, Supplementary Data 5B) GV genes exhibit no expression in the mock sample, with the exception of few genes ($n = 36$) that have non-zero TPM values (maximum = 0.29 TPM). This likely corresponds to traces of mRNA loaded in few particles not fully inactivated by heat treatment, although no sign of infection was detected 24 h pi.

As for the host, PCA analysis of the GV genes showed a tight clustering of the replicates, and a strong segregation of the different timepoints with the exception of T5 and T6 (Figure S4B). This suggests that GV gene expression is highly dynamic from 30 min to 6 h pi and

remains stable from 7 h pi until the end of the infectious cycle. Furthermore, as previously noticed for mimivirus[25], virocell mRNA population is dominated by viral transcripts by the end of the replication cycle, with 97.5% of the mapped RNA-seq reads originating from viral genes at T6 (Figure S2).

The clustering of viral genes by hkmeans (k = 5) revealed distinct patterns with significant enrichment of specific functions (Fig. 3A), and correlation with the presence of previously identified motifs in their promoters (Supplementary Data 1). No new motifs were identified using MEME-suite[41] and Homer[42]. The first cluster (GV-1) shows a robust expression from the beginning of the infection (30 min pi), followed by a gradual decline over time. The majority (60%, 148/243) and significant proportion of the genes (*P* value = 5.7 × 10⁻⁸, Supplementary Data 3C) have no known function, highlighting that most of the viral functions involved in the early stages of cell takeover are unknown. The rest are genes coding for Sel1 repeats-containing proteins, that are potentially involved in protein-protein and host interactions[43]. The second cluster (GV-2), with peak expression between 1h15min and 4 h pi, is also enriched in genes probably involved in protein-protein interactions (Ankyrin and FNIP repeats).

The third cluster (GV-3) contains all the genes involved in DNA replication and repair, such as the DNA polymerase, the PCNA sliding clamp and several copies of the small replication factor C (Supplementary Data 5B). It also reveals a strong activation of the virally-encoded transcription-related genes. Indeed, all of the 8 DNA-directed RNA polymerase subunits (RPB1-2, RPB5-7 and RPB9-11) are found within this cluster, as well as the mRNA capping enzyme, the poly(A) polymerase, the TATA-box binding protein (TBP) and 3 transcription factors (including TFIIB and TFIIS). Interestingly, mass spectroscopy (MS)-based proteomics of purified GV particles (see "Methods") revealed that RNA polymerase subunits are packaged in virions (Supplementary Data 6A). The same applies to the early TF (mchi_571), expressed late during the infectious cycle (cluster GV-5, see further), like in poxviruses[44]. This is in line with the discovery that RNA polymerase subunits and early TF proteins are present in the protein-shielded genomic fiber of mimivirus[45]. Such preloading allows for rapid initiation of transcription at subsequent infection[45]. The fourth cluster (GV-4), mostly expressed in T3-T4 (from 3h15min to 6 h pi) contains many zinc-finger domain proteins, as well as the VLTF3-like late TF (mchi_455), a core *Nucleocytoviricota* gene.

The largest and late expressing cluster (GV-5, from 5 h pi to the end of the infectious cycle) contains all the genes coding for the morphogenesis proteins, which comprises structural capsid proteins and the packaging ATPase[46]. As expected, this cluster strongly correlates with the proteins detected in GV virions by MS-based proteomics (Supplementary Data 6A and Fig. 3A). Genes coding for transmembrane-domain proteins are also enriched. Corresponding proteins are probably linked to the inner membrane layer found in viral particles (Fig. 3A) as they are predicted to localize at the endoplasmic reticulum by DeepLoc[47] (Figure S6), membranes of which are the source of *Megamimivirinae* virion inner membrane[48]. Genes coding for collagen proteins are also enriched in this cluster, in agreement with their localization at the surface of viral particles[49]. In addition, *Megamimivirinae* virions are surrounded by heavily glycosylated fibrils[50]. Accordingly, we found cluster GV-5 to be strongly enriched in genes involved in carbohydrate metabolism, which includes 6 out of the 8 encoded glycosyltransferases. Finally, the most expressed gene in this cluster is a long non-coding RNA gene (mchi_663) homologous to R549b in mimivirus[25].

Overall, megavirus chilensis exhibits expression profiles of key functions that are similar to mimivirus during the replication cycle, whether infecting *A. castellanii*[25] or *A. polyphaga*[40]. After 3 hours post infection, viral genes involved in DNA replication and transcription are highly expressed, and at the end of the infection cycle, genes

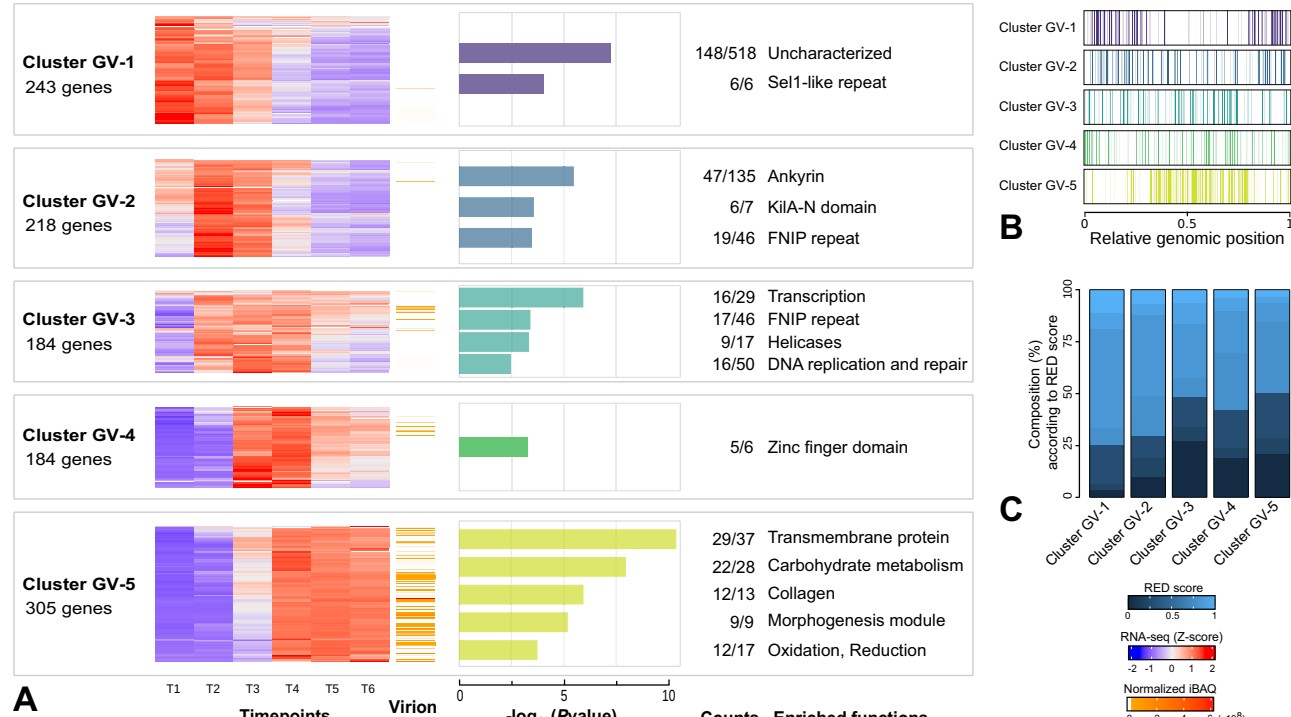

**Fig. 3 | Transcriptional patterns of megavirus chilensis genes during the infectious cycle. A** Hkmeans clustering of the GV genes with cluster names and number of genes in each cluster indicated on the left. The heatmaps show the expression patterns (Z-scores of log$_2$-transformed TPM values) of each gene (rows) along the different timepoints (columns), averaged over the replicates. Normalized iBAQ values of the proteins identified by MS-based proteomics in viral particles are also indicated. Colored histograms display one-sided hypergeometric test $P$values of the functional categories (see "Methods") significantly enriched in each cluster ($P$values ≤ 0.005). The numbers on the right indicate the number of genes with a given functional annotation (in cluster vs all expressed genes). **B** Relative genomic localization of all the GV genes assigned to each cluster. **C** Histogram of the proportion of GV genes according to their gene ancestry within the *Imitervirales* order (Figure S7). Gene age is measured by relative evolutionary divergence (RED) as defined by ref. 94 (see "Methods"). The oldest genes have a RED score of 0 and the most recent a RED score of 1.

associated with sugar metabolism, collagen, and capsid production are expressed in both viruses.

We next questioned the genomic distribution of the GV genes along the genome as a function of their expression timing. As shown in Fig. 3B, GV genes are not uniformly distributed, with gene density gradually shifting from high concentration of early-expressed genes at genomic extremities, to high concentration of late-expressed ones at the center of the genome. In addition, gene age is not equal between clusters. By computing the relative evolutionary divergence (RED) of megavirus chilensis genes based on their conservation within the *Imitervirales* order (see "Methods"), we found that the proportion of recently acquired genes is higher in early-expressed clusters and conversely ancient genes are more frequent when lately expressed (Fig. 3C). To schematize, our data support a model in which more recently acquired genes involved in virus-host interactions are expressed first from the extremities of the genome, and older ones, especially those involved in virion morphogenesis, are subsequently expressed from the center of the genome. Similar trends of unequal distribution of ancient and recently acquired genes have been observed in several different families of GVs[43,51–53], including pandoraviruses[54], suggesting a common constraint in genome evolution.

As previously described, the majority of host transcripts exhibit decreased expression levels during the late stages of infection. This includes genes with viral homologs, such as those involved in transcription, which are differentially expressed in both, the host (Fig. 2) and the GV (Fig. 3A). Focusing on shared transcription-related genes, we found that their expression levels usually overlap towards T2-T3, but while host gene expression drastically drops right after, the expression of virally-encoded homologs is generally maintained until the end of the infectious cycle (Figure S8). Assuming viral homologs preserve cellular functions, like the poxviruses-encoded DNA-dependent RNA polymerase[44], transcriptional capacity of the virocell might be maintained by GV compensation. Nevertheless, there are numerous examples in giant viruses of virally-encoded homologs that evolved distinct functions from their cellular counterparts[33,55]. Further studies on the megavirus-encoded transcriptional machinery components will thus be required to explore their role within the virocell during infection.

## Megavirus vitis transpoviron has no effect on the virocell transcriptome

In addition to the infection of *A. castellanii* cells with megavirus chilensis, we performed a similar experiment with megavirus chilensis associated with megavirus vitis transpoviron (from[16]) (C + GV + Tpv, Fig. 1A). The aim was to reveal the transcriptional program of Tpv genes, as well as its potential impact on the virocell transcriptome (C + GV, Fig. 1A).

The mapping of RNA-seq reads on the Tpv genome first confirmed that all predicted Tpv genes are transcribed, with some as early as T2 (1h15min to 2h30min pi) (Fig. 4A). Tpv transcription then drastically increases at T4 (5–6 h pi) until the end of the GV infectious cycle (Fig. 4B). Interestingly, the weakly expressed mvtv_1 gene, with a maximal expression of 2.7 TPM compared to the other Tpv genes (minimum = 12.1, maximum = 798), has an opposite expression profile with strong repression from T4 onwards (Fig. 4B). Examination of this genomic locus shows transcription from the opposite strand, possibly originating from the downstream neighboring gene (mvtv_2, Fig. 4A),

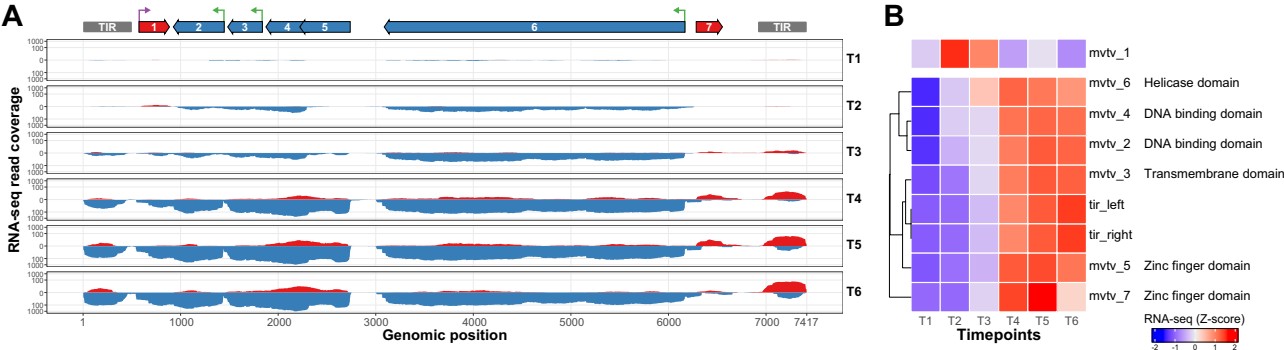

**Fig. 4 | Transcriptional patterns of megavirus vitis transpoviron genes during the infectious cycle. A** Coverage plot of RNA-seq reads (from one replicate) mapped using STAR[85] on the Tpv genomic sequence during GV infection. Coverage of reads from the forward strand is shown in red and reads from the reverse strand in blue. Tpv genome annotation is shown at the top with genes from the forward strand in red, reverse strand in blue, and TIRs in gray. Presence of motifs in promoter regions is depicted using purple and green arrows for early and late motifs, respectively. **B** Hkmeans clustering (k = 2) of the Tpv genes with associated dendrogram on the left. The heatmap shows the expression (Z-score of log2-transformed TPM values) of each gene (rows) along the different timepoints (columns), averaged over the replicates. Gene names and functional annotations are displayed on the right.

suggesting an antisense transcriptional interference. It is also the sole Tpv gene with an early regulatory motif[56] in its promoter region, located 54 nt upstream of the start codon (Fig. 4A and Supplementary Data 1).

Unexpectedly, we also observed transcriptional signal originating from Tpv TIRs as strong as in the predicted genes (Fig. 4A). TIRs are devoid of annotated protein-coding genes, but 3 short open reading frames (ORFs) of 33 to 51 amino acids were identified (Supplementary Data 7). No peptide from previously published MS-based proteomics data[16] of Vp-infected GV virions and purified Vp virions could be assigned to these ORFs. This suggests that TIR regions encompass highly expressed unidentified small proteins, or, most likely, ncRNAs of unknown function.

Comparison of the virocell with and without Tpv (C + GV+Tpv vs C + GV, Fig. 1A) revealed that only 4 (out of 12845) cellular genes are differentially expressed, with one weakly expressed and no particular function standing out (Supplementary Data 5A and Supplementary Data 8A). In addition, none of the GV genes are differentially expressed. In other words, Tpv has no significant impact on the virocell transcriptome.

To investigate potential Tpv and Vp integration into the GV genome, we sequenced using Nanopore long reads the genomic DNA of megavirus vitis, a closely related GV strain (97.9% average nucleotide identity with megavirus chilensis) from which zamilon vitis and megavirus vitis transpoviron were isolated[16]. We identified 12 megavirus vitis chimeric reads aligning to Tpv (Figure S9A), and 2 to Vp (Figure S9B), suggesting potential Tpv and Vp insertions within the megavirus genome. These insertions appear uniformly distributed throughout the genome (K–S test against uniform distribution Pvalue = 0.621), similar to observations in mimivirus[23]. These findings suggest potential GV diversification resulting from Tpv and Vp insertions. However, the low number of chimeric reads and their occurrence within essential genes (e.g., major capsid protein 3, mRNA capping enzyme, Figure S9A) indicate rare events probably often leading to evolutionary dead ends.

**Zamilon vitis virophage transiently modifies the megavirus chilensis transcriptome**

To further explore the transcriptome of this hyperparasitic system, we introduced the Vp by coinfecting *A. castellanii* cells with megavirus chilensis and zamilon vitis (C + GV+Vp, Fig. 1A). The experiment first revealed that all Vp genes are transcribed and fall into 4 clusters (Fig. 5A). For genes in the first cluster (Vp-1), a weak transcription signal

can be observed at T2, peaking at T3 and gradually decreasing onwards (Fig. 5A, Supplementary Data 5C). Among genes from this cluster is the DNA primase, probably involved in Vp DNA replication. Genes from cluster Vp-2 show steady expression from T3 to T6 and notably include the Vp-encoded integrase. The third and largest cluster (Vp-3) contains genes whose expression is delayed, peaking at T4. It includes all members of the morphogenesis module (minor and major capsid proteins, and the packaging ATPase), as well as 3 proteins sharing a similar fold (za3_1, za3_19 and za3_20) that are suspected to form spikes at the surface of the virophage capsid[57].

Finally, the latest expressed gene (za3_7, Fig. 5A), sole member of the Vp-4 cluster, encodes a transmembrane domain protein that is predicted to localize at the cell membrane and lysosome/vacuoles by DeepLoc (Supplementary Data 5C). Interestingly, according to our MS-based proteomics data (reprocessed from[16]), the protein is absent from purified Vp particles (Supplementary Data 9). By contrast, it is the most abundant Vp protein in GV particles when GV is infected by Vp (Supplementary Data 6B). Thus, this Vp-encoded protein is not associated with Vp virions, which lack internal membranes[58], but probably binds to inner membranes of GV virions.

Vp genes are expressed late during the GV infectious cycle, when the VF is operational, and are mainly controlled by GV-like late regulatory motifs (9 out 20 genes, Supplementary Data 1). However, akin to GV, it exhibits an organized gene expression pattern, with genes involved in DNA replication expressed first, followed by those involved in virion morphogenesis. This indicates that a hidden level of temporal gene regulation remains to be characterized.

To determine the Vp's impact on the virocell transcriptome, we next compared our transcriptomic data in the presence and absence of Vp (C + GV+Vp vs C + GV, Fig. 1A). Our analysis revealed a negligible impact of Vp on the host transcriptome, with only 6 cellular genes differentially expressed, 4 of which were weakly expressed (average expression < 5 TPM, Supplementary Data 5A and Supplementary Data 8A).

In contrast, Vp strongly disrupted GV gene expression, significantly altering the expression of 23% (263/1134) of its genes (Supplementary Data 8B). This substantial effect could be attributed to a bias arising from the introduction of a new partner with a finite pool of sequenced reads. As a control, we performed the same analysis excluding the Vp genome sequence from the mapping, i.e., only C and GV reference sequences were included. After confirming sufficient read coverage (Figure S3), we still found that 22% (254/1134) of GV genes were differentially expressed. Thus, the observed differential

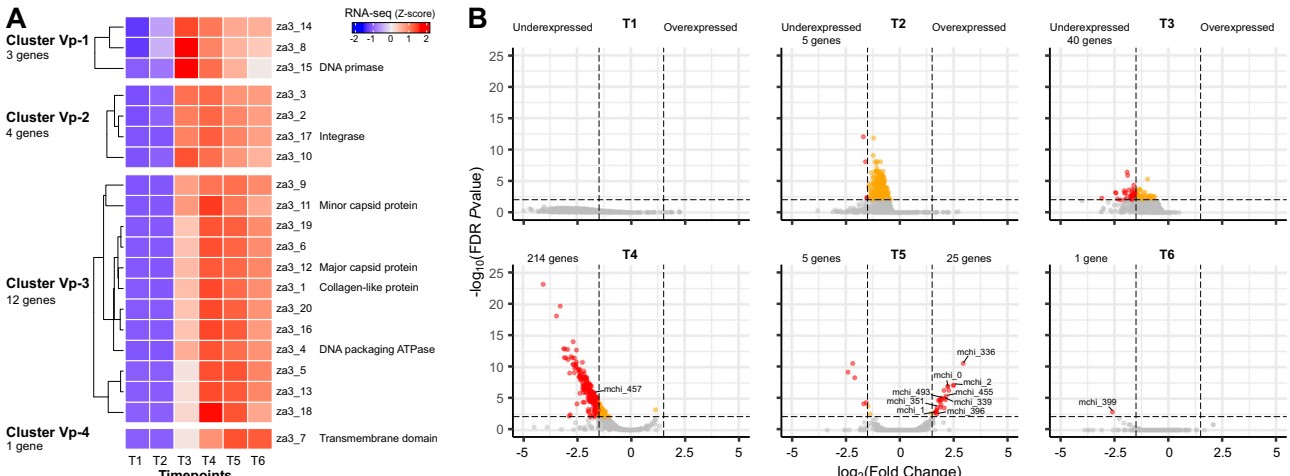

**Fig. 5 | Transcriptional patterns of zamilon vitis virophage genes during the infectious cycle. A** Hkmeans clustering (k = 4) of the VP genes with cluster names, number of genes in each cluster and associated dendrograms indicated on the left. The heatmap shows the expression patterns (Z-score of log₂-transformed TPM values) of each gene (rows) along the different timepoints (columns), averaged over the replicates. **B** Volcano plot of GV gene expression in the presence/absence

of the Vp (C + GV+Vp vs C + GV). Two-sided Wald test adjusted (Benjamini-Hochberg) FDR *P*values and fold change metrics were calculated using Deseq2[86]. Differentially expressed genes were identified using Deseq2 and EdgeR[87] with FDR *P*value < 0.01 and $|\log_2(FC) \geq 1.5|$. GV genes that passed the FDR *P*value and FC thresholds are shown in red, while those that only passed the FDR *P*value threshold are in orange. Source data are provided as a Source Data file.

expression of GV genes is indeed a result of its interaction with Vp, and not due to a bias in the proportions of mappable reads.

The effect of Vp on GV gene expression is mainly negative, as most differentially expressed GV genes (238/263) are underexpressed in their presence. This mainly occurs at T4 (Fig. 5B), at the same time as peak expression for most Vp genes (Fig. 5A). Competition for transcription machinery might thus occur between the two viruses (GV and Vp) at this time point. This is supported by the fact that Vp genes are globally more efficiently expressed than GV genes (Figure S10).

Since most of the underexpressed GV genes are expressed late (with 80% from cluster GV-5, Fig. 3A) and include important genes from the morphogenesis module, such as the major capsid protein (mchi_457, Fig. 5B and Supplementary Data 8B), one could expect that Vp coinfection alters GV particles protein composition. We thus performed MS-based analyses of GV virion in the presence and absence of Vp coinfection (see "Methods"). As shown in Supplementary Data 6C, none the virion-associated GV proteins exhibit differential abundance between the two conditions (with FDR *P*values < 0.05 and $|\log_2(FC)| \geq 1.5$ thresholds). This data nicely correlates with the fact that all of the underexpressed GV genes (with the exception of mchi_399) recover normal expression strength by the end of the infectious cycle (T6, Fig. 5B). Taken together, these data show that although Vp has a strong repression effect on GV transcriptome, it is only transitory and do not alter mature virion protein composition. Regardless, such transient changes might still have consequences on the speed of GV virion formation, and thus extend the period of time for mature Vp virions to be generated prior to host cell lysis. Further experiments will be needed in order to address such hypothesis. It is also possible that GV genes transcriptional level is sufficiently high that Vp-induced downregulation has no phenotypic effect on GV.

Not all differentially expressed GV genes are repressed in the presence of the Vp, being 25 of them upregulated at T5 (Fig. 5B and Supplementary Data 8B). Among them, 6 strikingly colocalize in GV TIRs, with 3 next to each other identical on each TIR: the mchi_0/mchi_1133 ncRNAs, the Bro-N domain-containing mchi_1/mchi_1132, and mchi_2/mchi_1131 that are homologous to za3_9 in Vp[12]. Other activated functions include protein folding with two chaperons (the DnaJ-like mchi_351 and the HSP70 mchi_493), and DNA interaction with the mchi_396 topoisomerase 2 and the MC1-domain containing

mchi_339. Interestingly, the latter is the most abundant GV protein in purified Vp particles (Supplementary Data 9). This suggests that this GV protein, recently proposed to be involved in mimivirus DNA compaction and packaging[59], has a similar function not only in megavirus GV but potentially in zamilon Vp as well.

Finally, the most upregulated gene in the context of virophage coinfection (log₂(FC) = 2.95, FDR *P*value = 2.5 × 10⁻¹¹, Supplementary Data 8B and Fig. 5B) is the mchi_336 transcription initiation factor (TFIIB). It is worth mentioning that this gene may be essential for GV replication. Indeed, knock-out (KO) by homologous recombination with a selection marker[60] resulted in a mixture of wild type and mutant viral particles. While mutants were rapidly outcompeted by wild-type viruses in the absence of selection, a complete loss of mutants was also observed with an increased number of passages under selection, indicating that mchi_336 KO is associated a high fitness cost. In addition to mchi_336, the GV-encoded mchi_455 late TF is also significantly upregulated by Vp coinfection (Fig. 5B). Altogether, this indicates that the Vp transiently activates key GV-encoded functions, likely to support its own gene expression and replication.

**Zamilon vitis virophage induces megavirus vitis transpoviron late gene overexpression**

Our previous transcriptomic comparisons highlighted the effects of Vp, and lack of effect of Tpv, on the virocell transcriptome. We next explored the reciprocal impact of Vp and Tpv on each other. To this end, we first compared the complete coinfection experiment (C + GV+Vp+Tpv, Fig. 1A) to the one excluding Tpv (C + GV+Vp, Fig. 1A), in order to reveal the potential effects of Tpv on the Vp transcriptome. None of the Vp genes passed the differential expression thresholds in this comparison (Supplementary Data 8C). Thus, not only Tpv has no effect on the virocell transcriptome, it has no effect on Vp genes expression either.

Reciprocally, we compared the full system (C + GV + Vp + Tpv, Fig. 1A) to the one without Vp (C + GV + Tpv, Fig. 1A), to decipher the effect of Vp on Tpv gene expression. We first observed a delay in Tpv transcription in the presence of Vp, with 3 Tpv genes (mvtv_2, mvtv_4 and mvtv_6) significantly underexpressed at T2 and/or T3 (Fig. 6 and Supplementary Data 8D). Importantly, in the C + GV + Tpv condition, Tpv is carried by the GV, while in the C + GV + Vp + Tpv it is brought

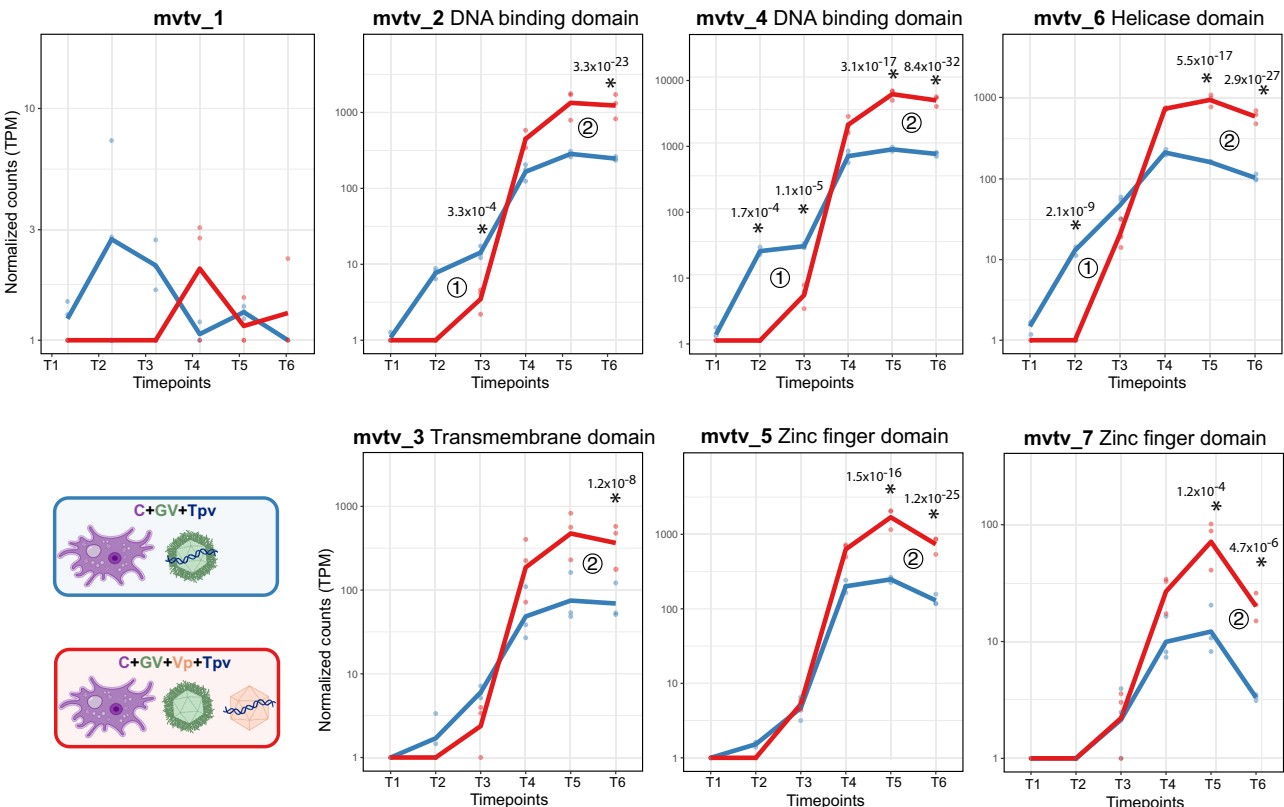

**Fig. 6 | Expression of megavirus vitis transpoviron genes in the presence or absence of the zamilon vitis virophage.** Shown is the normalized expression (TPM values) of all Tpv protein-coding genes across the GV infectious cycle in the presence (red) or absence (blue) of Vp. Stars indicate significant differential expression between the two conditions at a given timepoint with FDR $P$value < 0.01 and $|\log_2(FC)| \geq 1.5$ using Deseq2 and EdgeR. Two-sided Wald test adjusted (Benjamini-Hochberg) FDR $P$values are noted above each star. The numbers (1 and 2) indicate the two different effects detailed in the main text, with (1) for the delayed expression and (2) for the late overexpression of Tpv genes when Vp is present. Source data are provided as a Source Data file. Icons representing partners were created with BioRender.com.

along by the Vp virions (see "Methods" and Fig. 1A). The delay is therefore probably due to a difference in the accessibility of Tpv DNA for transcription, either because of delayed access to the transcription machinery or, most likely, because of a later opening of Vp virions.

Secondly and more importantly, all Tpv genes, except mvtv_1, are overexpressed in the presence of Vp at late timepoints (T5 and/or T6, Fig. 6). Thus, Vp induces a global increase of Tpv late gene expression. Since Vp depends on the GV transcription machinery[11,25] and does not encode TFs, we hypothesize that this upregulation of Tpv genes is not directly induced by the Vp, but rather indirectly through GV interaction. Indeed, as previously shown, the Vp upregulates the mchi_336 TFIIB and the mchi_455 late TF (Fig. 5B). The strong late global increase of Tpv gene expression might thus result from the transient Vp-induced upregulation of these GV-encoded TFs.

Together, these comparisons highlight an asymmetrical relationship between the two entities, with no effect of Tpv on Vp transcriptional program, but a strong global increase of Tpv expression indirectly induced by Vp via GV. Interactions between GV, Vp, and Tpv are therefore highly intricated at the transcriptional level.

## Discussion

This work provides a detailed picture of the transcriptional landscape of a complex quadripartite hyperparasitic system. If we first consider the GV single infection, the relative global expression of host genes considerably decreases during infection (Fig. 7A), while megavirus expression increases and dominates the transcriptome by the end of the infectious cycle (Figure S2), like in mimivirus[25]. However, specific

cellular functions are activated by viral infection, including transcription-related genes, highlighting that GVs with purely cytoplasmic infectious cycle can drastically reshape the host cell transcriptional program. GV transcription starts with a pre-loaded virally-encoded transcription machinery[45], but may therefore also depend on cellular machinery for subsequent stages of infection. The virally-encoded transcription machinery may contribute to intermediate and/or late gene expression as well, not only to early transcription. Proteins packaged in virion, such as the early TF, are usually lately expressed during infection (cluster GV-5, Fig. 3A), but viral RNA polymerase subunits are expressed much earlier (cluster GV-3, Fig. 3A), suggesting that they could play a role in all phases of infection. This raises the question of whether the two transcriptional apparatuses are truly totally independent and hermetically sealed, with one exclusively viral confined to the VF, and the cellular one in the nucleus. The transcription processes occurring in the virocell could also partly rely on chimeric virus-host RNA polymerase complexes. An in-depth study of the protein composition and cellular localization of viral and cellular components during infection will be necessary to explore this hypothesis.

Our study also revealed the timing of virophage and transpoviron gene expression during infection (Fig. 7A), as well as their intricate interactions with the virocell (Fig. 7B). As both are mainly controlled by late regulatory elements, their genes are expressed late when the GV-encoded TFs are available in the VF. However, the exact timing of Tpv genes expression may vary depending on the carrier (GV or Vp), which we hypothesize to be a consequence of late Vp opening. The Tpv does not alter GV and Vp gene expression, nor does it modify or disrupt

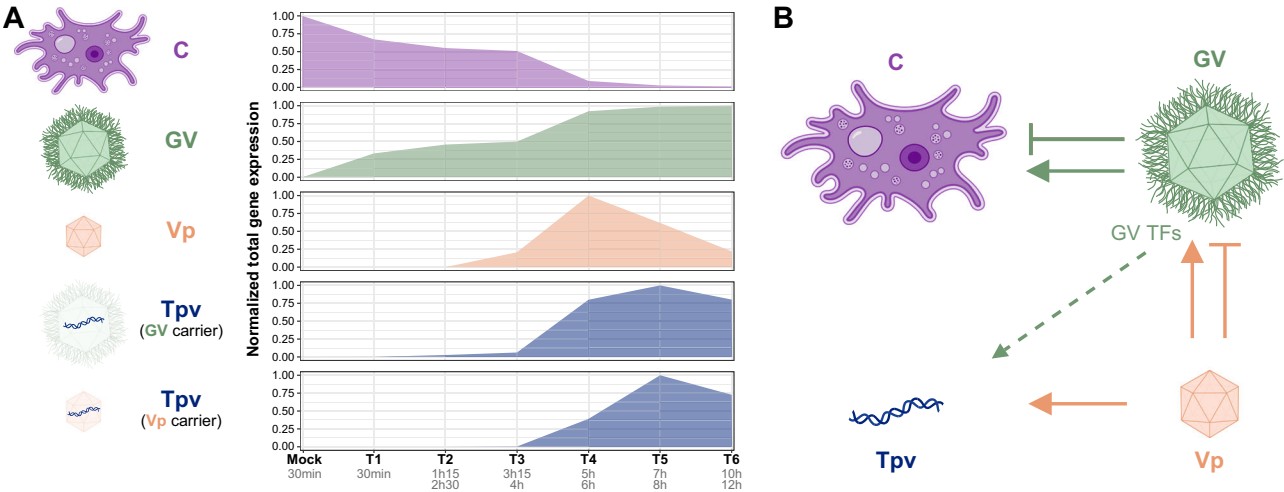

**Fig. 7 | Total gene expression and transcriptional responses to interactions of *A. castellanii*, megavirus chilensis, zamilon vitis and megavirus vitis transpoviron during infection. A** Total gene expression of each player (C, GV, Vp and Tpv) computed from the sum of TPM values at each timepoint, normalized by maximal total expression over the time course. For C and GV, the C + GV condition (Fig. 1A) was considered, for Vp, the C + GV+Vp condition was considered, and for Tpv, C + GV+Tpv and C + GV+Vp+Tpv conditions were considered. **B** Schematic diagram of the effect of the different players on the others' gene expressions. Arrows indicate induced over- (triangular arrow head) and under- (perpendicular arrow head) expression of the targeted player genes. Arrows are colored according to the player that induces this differential expression. Dashed arrow indicates a potential indirect upregulation of Tpv genes by giant virus-encoded TFs in presence of Vp. Source data are provided as a Source Data file. Icons representing partners were created with BioRender.com.

cellular functional pathways. It is therefore opportunistic, taking advantage of the VF for replication and the GV or Vp for propagation, confirming the commensalism interaction previously proposed[16]. However, the relationship between Vp and virocell is more complex than expected. Indeed, by hijacking the VF, it drastically alters the expression of late GV genes (Fig. 7B), probably by competing for the transcription machinery. However, this phenomenon is only transient and does not affect the protein composition of mature GV particles or their production[12,16]. We thus reveal that the zamilon vitis Vp is a transient hyperparasite of the megavirus chilensis GV. Even more surprising, the presence of Vp induces a global increase in Tpv transcription (Fig. 7B) that we hypothesize to be indirectly induced by GV-encoded TFs upregulation. So, even if this hyperparasitic system is at equilibrium, with no apparent phenotypic effect on the viral part, it conceals a complex network of interactions at the transcriptional level between the various players.

While this study focused on a specific system, GV-related hyperparasitic systems encompass a diverse array of virophages and cellular hosts[10,11,13], exhibiting varying degrees of positive, negative or neutral interactions. Other giant virus/virophage pairs, such as sputnik/mimivirus and mavirus/crov, demonstrate more pronounced parasitic relationships. It is tempting to speculate that the observed zamilon/megavirus transient hyperparasitic interaction might represent an evolutionary transition from strict parasitism towards commensalism[61]. This hypothesis is supported by experimental coevolution studies on mavirus and crov, showing increased virophage replication with decreased giant virus exploitation[62]. Similarly, the transpoviron commensal status confirmed in our study could reflect an adaptation from a more parasitic role. To fully understand the impact of the transpoviron and potential loss of virulence, analysis within a more parasitic system like mimivirus/sputnik would be necessary[10,23]. Regardless, as previously observed, both transpovirons and virophages[23], can integrate into GV genome (Figure S9), directly contributing to giant virus genomic diversity.

While the quadripartite system we analyzed in this work is complex, it represents only a portion of a larger ecosystem. *Acanthamoeba* hosts various symbionts and pathogens[63] that can interact with infecting giant viruses[64]. Not to mention that *Acanthamoeba* is by itself

an opportunistic pathogen of humans and animals. We are thus faced with a fascinating theater of interlocking parasites.

## Methods
### Virus production and purification
Megavirus chilensis (GV), megavirus vitis and zamilon vitis were produced and purified as described[16,65]. Megavirus chilensis containing the transpoviron and zamilon vitis free of transpoviron were obtained from coinfection experiments carried out previously[16]. Briefly, *A. castellanii* cells were coinfected with megavirus chilensis and zamilon vitis. After cell lysis, the culture was centrifuged at 10 000 g for 25 min to pellet the giant virus and the supernatant containing zamilon vitis (Vp + Tpv) was purified as described[16]. The pellet contained megavirus chilensis and zamilon vitis particles that stick to the giant virus. To separate them, the pellet was incubated in Tris buffer (40 mM pH 7.5) containing 250 mM DTT at 55 °C for 90 min. The resulting population of giant viruses was washed twice in Tris buffer and cloned[66]. Viral clones were screened by PCR to verify the presence of the transpoviron and the absence of virophage. One clone of megavirus chilensis containing the transpoviron (GV + Tpv) but free of virophage was recovered and amplified prior purification. To produce zamilon vitis free of transpoviron (Vp), purified particles of zamilon vitis were used to coinfect *A. castellanii* with the B-clade moumouvirus maliensis that previously showed poor efficiency to replicate a C-clade transpoviron[16]. The resulting population was cloned and clones were screened by PCR to isolate a sub-population devoid of transpoviron. The virophage particles devoid of transpoviron were separated from the giant virus by several steps of filtration/centrifugation and amplified with megavirus chilensis prior purification. All giant viruses and virophages preparations were controlled by negative staining observation after purification and the presence or absence of transpoviron was verified by PCR.

For mock-infections, viruses were inactivated by heat at 80 °C for 7 h.

### (co-)infections experimental setups
For transcriptome analysis, four different conditions were analyzed: *A. castellanii* strain Neff (ATCC 30010) cells infected with megavirus

chilensis (C + GV), or infected with megavirus chilensis containing the transpoviron (C + GV+Tpv), or infected with megavirus chilensis and the virophage (C + GV+Vp), or infected with megavirus chilensis and the virophage containing the transpoviron (C + GV+Vp +Tpv) (Fig. 1A).

For each condition, 15 T175 culture flasks containing proteose-peptone-yeast extract-glucose (PPYG) medium were inoculated with $10^5$ cells per $cm^2$, at 32 °C. Cells were infected (12 flasks) or mock-infected (3 flasks) at MOI 50 for the giant virus and a large excess of virophage (about 100 for 1 virus) when applicable. The concentration of transpoviron is unknown. After 30 min, inoculum was removed and cells were washed three times with 30 mL of PPYG to remove excess of viruses. Cells were harvested at 800 g for 10 min at different time-points from 30 min to 12 h post-infection to ensure comprehensive transcript capture throughout the infection (i.e 2 flasks at 30 min, 1 flask at 1 h 15 min, 2 h 30 min, 4 h, 5 h, 6 h, 7 h, 8 h, 10 h and 12 h and 2 flasks for mock-infected cells). Cell pellets were resuspended in 1.5 mL of RLT buffer complemented with β-mercapto-ethanol (Qiagen RNeasy midi kit), frozen in liquid nitrogen and stored at −80 °C until RNA extraction. One flask of mock-infected cells was kept at 32 °C for 24 h to verify the absence of infection for each condition. All experiments were performed in biological triplicate.

## RNA extraction and sequencing

For RNA extraction, in order to limit the number of samples for sequencing, pellets were pooled as follows to obtain 7 samples per infection condition: mock, T1 (30 min pi), T2 (1 h 15 min and 2 h 30 min pi), T3 (3h15 and 4 h pi), T4 (5 and 6 h pi), T5 (7 and 8 h pi) and T6 (10 and 12 h pi) (Fig. 1B). Total RNA was extracted for all 84 samples using the RNeasy midi kit (Qiagen) according to the instructions provided by the manufacturer. All samples were treated with Turbo-DNase (life technology) to eliminate DNA contamination and quantified with a Nanodrop spectrophotometer.

82 out of the 84 samples were successfully polyA-selected and sequenced by the Joint Genome Institute (JGI) with $2 \times 151$ bp paired-end reads using the Illumina NovaSeq S4 plateform (Supplementary Data 10).

## Reference genomes and annotations

Genomic sequences and gene annotations were gathered for all the partners. The *A. castellanii* Neff genome sequence assembly (GenBank assembly GCA_021020605.1) and corresponding gene annotations were obtained from (https://zenodo.org/records/5547275)[67]. Several genomic coordinates were manually corrected, including the BAESF_06046b Arginine--tRNA ligase (Supplementary Data 11). We also used the previously published genomes and gene annotations of the zamilon vitis virophage and megavirus vitis transpoviron (GenBank accessions MG807318.2 and MG807316.1, respectively)[16].

Additional annotations for the host genome were obtained using blastkoala (v3, eukaryotes database) and ghostkoala (v2, genus_prokaryotes, family_eukaryotes and viruses databases)[68]. In case of discrepancies, only blastkoala predictions were retained (Supplementary Data 5A). Protein annotation was supplemented by MMseqs2[69] matches to UniRef50 (query coverage > 50% and $E$value ≤ $10^{-5}$).

Transmembrane domains and cellular localization were predicted for GV, Vp, and Tpv proteins with Phobius[70] (transmembrane domain > 0.8) and Deeploc-2.0[47] (Supplementary Data 5B-D) with default parameters: high-quality model (slow) and long output format. For Deeploc-2.0 subcellular localization prediction, we applied recommended default cutoffs: cytoplasm (0.4761), nucleus (0.5014), extracellular (0.6173), cell membrane (0.5646), mitochondrion (0.6220), plastid (0.6395), endoplasmic reticulum (0.6090), lysosome/vacuole (0.5848), golgi apparatus (0.6494) and peroxisome (0.7364).

The genomic sequence of megavirus chilensis was reassembled to resolve terminal inverted repeats that were missing from the original genome sequence (Figure S1)[27]. We used a combination of 150-bp paired-end Illumina reads and Pacbio long reads from[71] using the Unicycler assembler v0.5.0[72] with default parameters. The resulting assembly graph was visualized using Bandage v0.9.0[73] to identify potential TIRs and polishing was subsequently applied using Pilon v1.23[74] with default parameters and the Illumina reads mapped with BWA mem v0.7.17[75]. We next performed gene annotation by transferring previous annotations (from accession JN258408.1) using Flo[76], and gene prediction using GeneMark[77] with the --virus option complemented with Funannotate (https://github.com/nextgenusfs/funannotate) supported with our RNA-seq data (JGI project ID 1287923, C + GV samples, Supplementary Data 10). We also used a combination of tRNAmod[78] and tRNAscan-SE[79] for transfer RNAs annotation. Finally, a manual curation was performed to precisely define UTRs and non-coding genes. Functional annotation was done by homology search against the NR (release 05/17/2022), SwissProt (release 05/09/2022) and VOG43 (Virus Orthologous Groups, VOG2016) databases using BlastP v2.12 ($E$value < $10^{-2}$) and domain identification with InterProScan (v5.65-96.0)[80] and CD search[81]. In cases of overlapping domains, we used the preposition with, while the and preposition was used for distinct domains. The SignalP[82], Phobius[70], DeepLoc-1.0 and DeepLoc-2.0[47] algorithms were also applied to proteins without identifiable domain or homologue. The GenBank accession JN258408.2 has been updated according to the new genome sequence and annotations.

## RNA-seq data processing and analysis

Quality control was performed by the JGI using BBTools (v38.90, https://sourceforge.net/projects/bbmap/) with the following parameters: rqc.filter2.sh rna = t minlength = 49 qtrim = r maq = 10 trimq = 6 trimfragadapter = t phix = t maxns = 1 mlf = 0.33 removeribo = t entropy = 0.44 entropyk = 2 entropywindow = 40 removehuman = t removedog = t removecat = t removemouse = t khist = t removemicrobes = t sketch kapa = t clumpify = t tmpdir = null barcodefilter = f trimpolyg = 5. Resulting cleaned fastq files were quality-checked with Fastqc v0.11.9 (https://www.bioinformatics.babraham.ac.uk/projects/fastqc/). Out of the 84 RNA samples (7 time points, 4 conditions, and 3 biological replicates) all were successfully sequenced and passed quality control (Supplementary Data 10) with the exception of two samples (C + GV+Vp T4 replicate 3 and C + GV+Vp+Tpv T2 replicate 2, Supplementary Data 10).

Reference genomic sequence combining all partners present in each condition and corresponding annotation in GTF format were used with RSEM v1.3.3[83] using options –polyA and –bowtie2 (v2.4.4)[84], to obtain gene level quantifications. Normalized counts (TPM, Transcripts Per Million) were extracted from the count matrix generated by RSEM. Genes were considered significantly expressed and kept for following analyses when: expression ≥1 TPM in ≥ 2 replicates per timepoint, in ≥ 2 timepoints, in ≥ 1 condition.

Genome mapping of RNA-seq reads was performed to identify transcription for unannotated regions using STAR (v2.7.6)[85] (options --genomeSAindexNbases 11, --twopassMode Basic, --alignIntronMax 5000, --alignMatesGapMax 5000).

Differential expression analyses were done either between two timepoints in a single condition, or at the same timepoint between two different conditions, using DESeq2 v1.40.2[86] and edgeR v3.42.4[87]. Genes were considered differentially expressed when FDR $P$value < 0.01 and $|\log_2(FC)| ≥ 1.5$ by both methods.

Heatmaps were drawn for the different partners based on Z-score of $\log_2$-transformed TPM values averaged over the replicates and visualized using ComplexHeatmaps v2.16[88]. Genes were clustered using hkmeans factoextra v1.0.7 (https://CRAN.R-project.org/

package=factoextra), with hc.metric = euclidean, hc.method = ward.D2 and km.algorithm = MacQueen.

To ensure that sequencing depth was sufficient for all partners we performed saturation curves as proposed in ref.[89] for all conditions (Figure S3). Briefly, we randomly sampled the bam files resulting from read mapping from 0.01% to 100% of the data, and considered the number of genes of each partner as detected when covered by at least 5 read pairs as described in ref.[90]. This was performed from the three replicates at T4, as all partners had genes expressed at this time-point (Fig. 7A).

## Functional enrichment analysis

*A. castellanii* GO terms were obtained from https://zenodo.org/records/5547275[67] and used to compute functional enrichment by the topGO package v2.52.0 (https://bioconductor.org/packages/topGO) with Fisher exact test, weight algorithm and biological process ontology. GO terms were considered significantly enriched in a specific cluster compared to all the differentially expressed genes when $P$value ≤ 0.005.

For GV functional enrichment analyses, megavirus chilensis genes were manually classified in 38 functional categories (Supplementary Data 5B). Significant enrichment ($P$value ≤ 0.005) of functional categories were done using hypergeometric tests for each cluster (compared to all expressed genes).

## Motif enrichment in promoter sequences

Genomic sequences around TSS (250 nt upstream to 250 nt downstream) were extracted for all the 12845 expressed cellular genes. The FindMotif.pl script from the Homer package (v4.11.1)[42] was used to identify enriched motifs of length 6 to 12 nt (with options -len 6, 7, 8, 9, 10, 11, 12 -S50) in sequences from each cluster, using sequences not within cluster as background. Next, identified motifs were filtered to remove similar motifs and motifs with $P$values > $10^{-15}$ using the compareMotif.pl script. Motif density around TSSs was computed using the annotatePeaks.pl script with -size 500 -hist 1 option and values were averaged with a roll mean (window=25). The known motifs similar to identified motifs were found using the compareMotif.pl script.

For GV, Vp and Tpv, known *Megamimivirinae* regulatory motifs were searched in promoter regions, defined as 100 nt upstream to start codon. The early regulatory motif[56] with consensus AAAATTGA sequence was searched using Fuzznuc from the EMBOSS package v6.6[91] with default parameters and the late regulatory motif[25] with consensus sequence [AT](8)T[AC]TN(4)[AT](5)[AG]TA[TG]A was searched the same way but allowing 1 mismatch.

## Relative evolutionary divergence of megavirus chilensis genes

A rooted species tree was computed from *Imitervirales* genomes using OrthoFinder (v2.5.5)[92] with the diamond_ultra_sens option (Figure S7). The following GenBank accessions were used: *Oceanusvirus kaneohense* (KY322437.1), *Tethysvirus hollandense* (KC662249.1), *Tethysvirus raunefjordenense* (KT820662.1), *Rheavirus sinusmexicani* (GU244497.1), *Fadolivirus algeromassiliense* (MT418680.1), *Theiavirus salishense* (MF782455.1), *Yasminevirus saudimassiliense* (UPSH00000000.1), *Cotonvirus japonicum* (AP024483.1), *Megavirus baoshanense* (MH046811.2), *Megavirus chilense* (JN258408.2), *Megavirus powaiense* (KU877344.1), *Mimivirus bradfordmassiliense* (HQ336222.2), *Mimivirus lagoaense* (KM982402.1), *Moumouvirus australiense* (MG807320.1), *Moumouvirus goulettemassiliense* (KC008572.1), *Moumouvirus moumou* (JX962719.1), *Tupanvirus altamarinense* (MF405918.2), *Tupanvirus salinum* (KY523104.2), *Kratosvirus quantuckense* (KJ645900.1). The obtained phylogenetic tree was consistent with the previously published phylogeny[93]. We next computed relative evolutionary divergence (RED) scores[94] using the get_reds function from the Castor R package[95].

## Potential integration of megavirus vitis transpoviron and zamilon vitis in the megavirus vitis genome

Megavirus vitis genomic DNA was extracted from $5 \times 10^9$ purified viral particles using the PureLink™ Genomic DNA mini kit (Invitrogen) according to the manufacturer's protocol. Purified DNA was sent to Nanopore for sequencing. Megavirus vitis transpoviron and zamilon vitis integration events were searched from Nanopore long sequencing reads of megavirus vitis infected by zamilon vitis and megavirus vitis transpoviron (accession PRJNA1144910). Chimeric reads with one extremity aligning to the megavirus vitis genome (GenBank accession MG807319.1) and the other to megavirus vitis transpoviron or zamilon vitis were identified using BlastN (Evalue threshold < $10^{-50}$ and percentage identity > 70%).

## MS-based proteomics

For Vp virion proteome characterization, we reprocessed data from[16], with two biological replicates of Vp virions purified from *A. castellanii* cells coinfected with GV, Vp and Tpv. For analysis of GV virion proteome, we prepared and analyzed three biological replicates of GV virions purified from *A. castellanii* cells with or without coinfection with Vp. Proteins were solubilized in Laemmli buffer and stacked in the top of a 4–12% NuPAGE gel (Invitrogen). After staining with R-250 Coomassie Blue (Biorad), proteins were digested in-gel using trypsin (modified, sequencing purity, Promega) as previously described[96], except that Tris(2-carboxyethyl)phosphine hydrochloride was used instead of dithiothreitol. The resulting peptides were analyzed by online nanoliquid chromatography coupled to MS/MS (UHPLC Vanquish Neo and Orbitrap Ascend Tribrid, Thermo Fisher Scientific). For this purpose, the peptides were sampled on a precolumn (300 μm × 5 mm PepMap C18, Thermo Scientific) and separated in a 75 μm × 250 mm C18 column (Aurora Generation 3, 1.7 μm, IonOpticks) using a 80 min gradient. The mass spectrometry proteomics data have been deposited to the ProteomeXchange Consortium via the PRIDE[97] partner repository under the accession PXD052049.

Peptides and proteins were identified by Mascot (v2.8.0, Matrix Science) through concomitant searches against a reference database containing C, GV, Vp and Tpv sequences as well as classical contaminant (keratins, trypsin, etc. 250 sequences). Trypsin/P was chosen as the enzyme and two missed cleavages were allowed. Precursor and fragment mass error tolerances were set respectively at 10 and 20 ppm. Peptide modifications allowed during the search were Carbamidomethyl (C, fixed), Acetyl (Protein N-term, variable), and Oxidation (M, variable). The Proline software[98] (v2.2) was used for the compilation, grouping and filtering of the results (conservation of rank 1 peptides, peptide length ≥ 6 amino acids, false discovery rate of peptide-spectrum-match identifications < 1%[99], and minimum of one specific peptide per identified protein group). Proline was then used to perform a MS1-based label-free quantification of the identified protein groups based on specific and razor peptides, with the cross-assignment option enabled. Proteins identified in the contaminant database were discarded. Intensity-based absolute quantification (iBAQ) values[100] were computed for each protein in each sample. For Vp virion proteome, only proteins detected in the two biological replicates were considered. For GV virion proteome, only proteins identified by MS/MS in a minimum of two biological replicates and detected in the three biological replicates of one condition were considered. Final iBAQ values for each protein in each sample type were obtained by averaging iBAQ values normalized by the sum of iBAQ values in each replicate.

For a detailed comparison of GV virion proteome prepared from *A. castellanii* cells coinfected or not with Vp, a statistical analysis was performed using the Prostar software (v1.34.5)[101]. After $\log_2$ transformation, abundance values were normalized on the abundance value of GV major capsid protein before missing value imputation (SLSA

algorithm for partially observed values in the condition and Det-Quantile algorithm for totally absent values in the condition). Statistical testing was conducted with limma before *P*values adjustment using the Benjamini-Hochberg procedure. Differentially expressed proteins were selected using a $\log_2$(Fold Change) $\geq 1.5$ and an adjusted *P*value < 0.05.

## Knock-out of the GV-encoded mchi_336 gene

Knock-out of mchi_336 was performed by homologous recombination as previously described[60]. Briefly, homology arms were amplified from megavirus chilensis by PCR using the primers below:

    5'arm_F: cttttgcaaaaagcttGTGATCGCATTAAATATATTGAT
    5'arm_R: aattgctaatattttTCTGATAAATTATGATGACGAG
    3'arm_F: aaatagtcctttagaATCGATTGCAAGATTGATCG
    3'arm_R: cttatcgctgcggccgcTCGTTTTTATGTATTTCTTCTTT

The 350 base pairs homology arms were inserted into the HindIII and NotI sites of the vHB47 plasmid by InFusion (Takara). The plasmid was digested with HindIII and NotI (NEB) and *A. castellanii* cells were transfected with 6 µg of the linearized vector using polyfect (Qiagen). Infection was performed 1 h post-infection with megavirus chilensis and genomic integration of the selection cassette was verified. Cloning of the mutant viruses was performed according to the previously described protocol[60], leading to a total loss of mutant viral particles after 4 rounds of infection under selection with Nourseothricin.

## Statistics and reproducibility

No statistical method was used to predetermine the sample size. The validity of sample size was estimated post hoc based on saturation curves of the number of detected transcripts as a function of the percentage of RNA-seq data. Out of 84 samples, 2 were excluded from the analyses as RNA sequencing failed for these samples. All experiments were performed in biological triplicates. The experiments were not randomized. The investigators were not blinded to allocation during experiments and outcome assessment.

## Figures design

All graphics were created using R and icons representing partners in Fig. 1, Fig. 6, Fig. 7 and Figure S3 were created in BioRender. Bessenay, A. (2024) https://biorender.com/a57l329.

## Reporting summary

Further information on research design is available in the Nature Portfolio Reporting Summary linked to this article.

## Data availability

The raw and QC filtered RNA-seq data generated in this study are available from the Joint Genome Institute portal under the proposal ID 505029 [https://doi.org/10.46936/10.25585/60001241] with project IDs: 1287919 (SP 1287923) [https://genome.jgi.doe.gov/portal/Megchinscriptome_2_FD/Megchinscriptome_2_FD.info.html], 1248764 (SP 1248768) [https://genome.jgi.doe.gov/portal/Megchinscriptome_FD/Megchinscriptome_FD.info.html], 1287916 (SP 1287922) [https://genome.jgi.doe.gov/portal/Cafroenscriptome_2_FD/Cafroenscriptome_2_FD.info.html]. In addition, all accession numbers and sample IDs are available from Supplementary Data 10. Proteomics data generated in this study are available from the PRIDE database under the accession PXD052049. Nanopore long read sequences of megavirus vitis generated in this study are available from the SRA portal under the accession PRJNA1144910. Source data are provided with this paper.

## Code availability

Custom script codes used in this study can be accessed here: https://doi.org/10.6084/m9.figshare.25880140.

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

## Acknowledgements

The work (proposal 505029, https://doi.org/10.46936/10.25585/60001241) conducted by the U.S. Department of Energy Joint Genome Institute (https://ror.org/04xm1d337), a DOE Office of Science User Facility, is supported by the Office of Science of the U.S. Department of Energy under Contract No. DE-AC02-05CH11231 to C.A. and J.M.C. We thank Matthias Fischer for his support on the proposal. This work was also supported by the French National Research Agency (ANR-20-CE11-0001) to S.J. The proteomic experiments were also partially supported by the French National Research Agency under projects ProFI (Proteomics French Infrastructure, ANR-10-INBS-08) to Y.C. and GRAL, a program from the Chemistry Biology Health (CBH) Graduate School of University Grenoble Alpes (ANR-17-EURE-0003) to Y.C. This work was also supported by a doctoral fellowship obtained from Aix-Marseille University to A.B. We thank the PACA Bioinfo platform for computing support. We also acknowledge the support of Jean-Marie Alempic for the gene knock-out experiments and Sofia Rigou for carefully reading the manuscript.

## Author contributions

Conceptualization: S.J., C.A., J.M.C, and M.L. Formal analysis: A.B., S.J. and M.L. Investigation: A.B., L.B., Y.C., L.B., S.J, and M.L. Writing– Original Draft: A. B., S.J. and M.L. Writing–Review and editing: A.B., H.B., Y.C., J.M.C., C.A., S.J. and M.L. Visualization: A.B. and M.L. Input on analysis: C.A., H.B. and J.M.C. Funding Acquisition: S.J., C.A. and J.M.C. Supervision: S.J. and M.L.

## Competing interests

The authors declare no competing interests.
