## [Peer Review File · Nature Communications]

Complex transcriptional regulations of a hyperparasitic quadripartite system in giant viruses infecting protistsREVIEWER COMMENTS

Reviewer #1 (Remarks to the Author):

This study asks how giant viruses interact with their host and their parasites - virophages and transpovirons - at the level of gene expression and regulation. The transcriptional dynamics of all four partners was investigated in co-infection experiments using time-resolved RNA-sequencing, and megavirus chilensis, its virophage zamilon vitis, the megavirus vitis transpoviron and the amoeba host *Acanthamoeba castellanii* as model system. The authors describe the intricate gene regulatory interplay between the virocell, virophage, and transpovirons. They found that the transpoviron represents a commensalistic element without impact on gene expression of host, giant virus, and virophage. Conversely, the virophage strongly reshaped the virocell transcriptome, yet only transiently and without an apparent influence on giant virus protein expression and particle formation.

This study is a beautiful RNA-sequencing based study with an elegant experimental set-up, supplemented by proteomic and genetic analysis to further investigate gene/protein function. The manuscript is well-structured and excels through thorough and very well-done data analyses, data visualization, interpretation and critical discussion. It represents an important contribution to the field and beyond.

I thus only have very few minor comments:

1. The manuscript title lacks a reference to viruses/protists
2. Subtitles and (supplementary) Figure titles are mostly very broad statements that generalize the observations in this study to all hosts/giant viruses. To enhance clarity, the titles should be more specific and include both the host and the GV name, for instance "The *Acanthamoeba* transcriptome is reprogrammed by megavirus chilensis infection" instead of "The host cell transcriptome is reprogrammed by giant virus infection".
3. Replicates or repeats: "All experiments were performed in triplicate." (line 500). Please clarify, whether the replicates were performed at the same time in parallel, i.e. using the same amoeba and virus stocks and thus resembling technical replicates, or whether the experiments have been repeated at different dates with different amoeba cultures (meaning they are repeats, or biological replicates).
4. Is there a potential that the polyA enrichment used introduces any bias?
5. RNA from temporarily close time points of the infection experiments were pooled before RNA sequencing. While the number of time points analyzed is sufficient and actually exceeds the number of time points in similar previous studies, it would still be interesting to understand the rationale for this. Was this done to increase total RNA amounts per sequencing library, or was there another reason to abstain from the higher temporal resolution?
6. How was the RED score calculated for Figure 3C? As the RED score is dependent on a phylogenetic tree, which tree was used? Please add details to the methods section.
7. L. 342 Supplementary Table 11 does not exist. Should likely read Supplementary Table 8B.

Reviewer #2 (Remarks to the Author):

The study investigates the transcriptional interplay among four components in a complex host-virus system: a host (*Acanthamoeba castellanii*), a giant virus (megavirus chilensis), a virophage, and a transpoviron. The aim is to decipher their transcriptional programs and interactions within the hyperparasitic system. In addition, proteomic data of the purified virions was also generated.

I think the manuscript is important as it explores the transcriptional complexity of hyperparasitism in a protist host, which will provide important baseline data and insights for future studies on hyperparasitism in diverse eukaryotes. I do not have any major concerns regarding the methods, analysis and interpretation. However, I believe some additional discussion is warranted regarding the transcriptional reprogramming of *Acanthamoeba* (see my comment below), along with some additional details in the method section.

Please see my comments below:

Regarding Method - Virus production and purification: Understandable that the authors have used the previously described method. However, a brief description of the approach would help the readers of the manuscript who are mostly interested in the results and want some methodological context. I encourage adding some more details in these methods.

Line 494: What was the MOI for? Is this MOI value specific to megavirus? In what concentration the viroplasm and transpoviron were added? How was the inoculum concentration of transpoviron measured?

Line 524,525: Was there a quality or confidence score cut-off that was used for Deeploc-2.0? Other parameters of the program should also be specified.

Regarding the transcriptional reprogramming of *Acanthamoeba* in response to giant virus infection - the authors should provide additional discussion to compare and contrast the response they saw in their study with the previous transcriptional studies - particularly members of Mimivirales groups and their hosts. For example, I believe one or more previous studies exist on transcription profile of *Acanthamoeba*-Mimivirus system:

<https://www.ncbi.nlm.nih.gov/pmc/articles/PMC2860168/>
<https://www.biorxiv.org/content/10.1101/2022.07.20.500700v2>

Line 257-259: I believe this has also been suggested in several more studies. See:

[https://www.cell.com/trends/genetics/abstract/S0168-9525\(06\)00367-2?large_figure=true](https://www.cell.com/trends/genetics/abstract/S0168-9525(06)00367-2?large_figure=true)

<https://www.sciencedirect.com/science/article/pii/S0042682214002992>

Line 267-270: As per the data, it seems like the vast majority of host transcripts were suppressed during the late stage of infection and viral RNA dominated the virocell, not only the ones that have homologs in the virus. Since the host transcript had a global suppression, I am not sure if a causal connection between the host and virus homologs can be made - the suggestion here is that the viral homologs possibly 'replaced' the functions of the host homolog - but we can not rule out the possibility that the viral homologs had a function that is distinct from the host counterparts. Further discussion could be added to clarify this.

Line 301: 'peculiar' or 'particular'?

General comment: Please ensure scientific names are italicized throughout the manuscript. For example, "*Acanthamoeba*".

Reviewer #2 (Remarks on code availability):

I believe the script used to analyze the data is made available in figshare, although I didn't find a readme with further instructions on how to use these scripts.

Reviewer #3 (Remarks to the Author):

This manuscript by Bessenay et al investigated the outcome of an evolutionary interplay of amoeba, *Acanthamoeba castellanii*, its giant virus, megavirus chilensis with its virophage, Zamilonvitis, and a transpoviron (a genetic parasite) known as megavirus vitis transpoviron. Some giant viruses, such as the megavirus vitis used in this study, carry linear extrachromosomal elements of about 7 kb coding for eight genes. They are known as transpovirons. Virophage and transpoviron are dependent on giant viral transcription machinery.

The authors have carried out extensive transcriptional analysis (RNA sequencing) at various time points during the infection to assess the outcome of the interaction of these four biological entities during a single infection cycle (up to 12 h, in triplicates). They show that, as expected, giant viruses significantly modify the host transcription profile. Interestingly, while coinfection with virophage resulted in transient changes in the giant viral transcriptome with no significant effect on either the fitness of the giant virus (giant virus production from viral factories) or the protein composition of the assembled giant viral particles, it led to the "overexpression" of the transpoviron genes. Just 30 min post-infection, a significant change was observed in the expression levels of host transcriptome with enhanced expression of Rho family GTPases, which could be involved in the host cytoskeleton remodeling. They have also observed a lowered gene expression in the host cell encystment. It appears that the presence of virophage enhances the expression of Tpv genes.

The experimental strategy is comprehensive and gives some insights into the changes in the transcriptional profile in a four-way hyperparasitic system and when one or more players are absent.

While most of these insights are, at best, confirmatory to what has been already known, the study lacks mechanistic or novel insights. Some hypothesis and follow-up experiments (long-term or otherwise) would have helped in gaining those insights. While the evolutionary outcome of a four-way host-nested parasite is complex and probably context-dependent, any generalizations (some hypotheses) would have added more value to this study. For example, it is well-known that host-parasite interactions are major drivers of diversity. Any insights on how or whether this four-level parasitism could lead to the diversification of giant viruses, or some thoughts on the loss of virulence and its consequence (if my interpretation of the data is correct) of Tpv would have greatly enhanced the manuscript.

Minor comments

- The Results and Discussion section can be just "Results" and the "Conclusion" is too long, it can be the "Discussion" section as the findings from the study have been captured with good context.
- In the Abstract, the authors have mentioned nested parasitism, but in the introduction, they talk about hyperparasitism. These terminologies can be a bit confusing, especially to the uninitiated in the host-parasite coevolution. I think there is a subtle difference in the concepts and the authors may clarify them in the introduction.
- Lines 255-259: The presence of the so-called "core genes" towards the middle of the genome has also been shown in all NCLDVs by a 2016 paper (PMID: 29308275) that can also be cited to support the observation, in addition to reference 51.
- The sentence (338-341), "As for Tpv coinfection, Vp effect on the host transcriptome is negligible, with only six genes differentially expressed, of which four are weakly expressed (average expression < 5 TPM, Supplementary Table 5A and Supplementary Table 8A)" can go towards the end of the same paragraph for better readability.
- Lines 398-405: The delay in the expression of some Tpv genes in the presence of Vp is a good insight from the study. It is not clear whether there is also a concomitant delay in the replication of the transpoviron in the presence of Vp. Also, in the absence of Vp, how many copies of transpovirons are made, and how the copy number (a determinant of its fitness) is affected in the presence of Vp.
- The selective packaging of transpoviron (again if my understanding is correct) into Vp rather than GV is interesting indeed, but it has to be experimentally confirmed. Can there be a scenario where transpoviron is present in both GV as well as Vp? If yes, can this also be considered?

Reviewer #3 (Remarks on code availability):

The authors have used standard tools to analyze their RNA sequencing data. I don't think created any new code.

Manuscript NCOMMS-24-30805- Point-by-point response

Reviewers' comments are in red, our responses to the comments in black and citations of the corrected manuscript in green.

New title: **Complex transcriptional regulations of a hyperparasitic quadripartite system in giant viruses infecting protists**

Reviewer # 1

This study asks how giant viruses interact with their host and their parasites - virophages and transpovirons - at the level of gene expression and regulation. The transcriptional dynamics of all four partners was investigated in co-infection experiments using time-resolved RNA-sequencing, and megavirus chilensis, its virophage zamilon vitis, the megavirus vitis transpoviron and the amoeba host Acanthamoeba castellanii as model system. The authors describe the intricate gene regulatory interplay between the virocell, virophage, and transpovirons. They found that the transpoviron represents a commensalistic element without impact on gene expression of host, giant virus, and virophage. Conversely, the virophage strongly reshaped the virocell transcriptome, yet only transiently and without an apparent influence on giant virus protein expression and particle formation.

This study is a beautiful RNA-sequencing based study with an elegant experimental set-up, supplemented by proteomic and genetic analysis to further investigate gene/protein function.

The manuscript is well-structured and excels through thorough and very well-done data analyses, data visualization, interpretation and critical discussion. It represents an important contribution to the field and beyond.

I thus only have very few minor comments:

We appreciate your kind comments on our work.

R1.1

The manuscript title lacks a reference to viruses/protists

We have changed the manuscript title to add reference to viruses and protists. It now reads: "Complex transcriptional regulations of a hyperparasitic quadripartite system in giant viruses infecting protists".

R1.2

Subtitles and (supplementary) Figure titles are mostly very broad statements that generalize the observations in this study to all hosts/giant viruses. To enhance clarity, the titles should be more specific and include both the host and the GV name, for instance "The Acanthamoeba transcriptome is reprogrammed by megavirus chilensis infection" instead of "The host cell transcriptome is reprogrammed by giant virus infection".

We have modified subtitles, supplementary figures and tables titles as requested.

R1.3

Replicates or repeats: “All experiments were performed in triplicate.” (line 500). Please clarify, whether the replicates were performed at the same time in parallel, i.e. using the same amoeba and virus stocks and thus resembling technical replicates, or whether the experiments have been repeated at different dates with different amoeba cultures (meaning they are repeats, or biological replicates).

Replicates were performed using distinct amoeba cultures with at least one week between 2 experiments, while maintaining the same virus stocks. Thus, they correspond to genuine biological replicates. The term “triplicate” has been replaced by “biological triplicate” throughout the text to avoid ambiguity.

R1.4

Is there a potential that the polyA enrichment used introduces any bias?

Poly(A) enrichment is a standard and widely used method for RNA-seq analysis of eukaryotic cells. A potential bias could arise if viral transcripts, unlike those of *Acanthamoeba*, lacked poly(A) tails. However, previous studies (Byrne et al. 2009, <https://doi.org/10.1101%2Fgr.091561.109> and Priet et al. 2015, <http://dx.doi.org/10.1093/nar/gkv224>) have demonstrated that Mimiviridae mRNAs exhibit poly(A)-tails and that megavirus chilensis poly(A)-polymerase is highly efficient, producing very long polyA tails. Furthermore, since both virophage and transpoviron use the viral transcription machinery, we are confident that no bias was introduced.

R1.5

RNA from temporarily close time points of the infection experiments were pooled before RNA sequencing. While the number of time points analyzed is sufficient and actually exceeds the number of time points in similar previous studies, it would still be interesting to understand the rationale for this. Was this done to increase total RNA amounts per sequencing library, or was there another reason to abstain from the higher temporal resolution?

To minimize experimental costs, we were constrained by the number of samples to sequence. We decided to sample all over the infectious cycle at frequent intervals, and to pool adjacent timepoints. This ensured that no transcript was missed during the infectious cycle.

We modified the text as follows:

“Cells were harvested at 800g for 10min at different timepoints from 30 min to 12h post-infection to ensure comprehensive transcript capture throughout the infection (i.e 2 flasks at 30 min, 1 flask at 1h15, 2h30, 4h, 5h, 6h, 7h, 8h, 10h and 12h and 2 flasks for mock-infected cells).”

and

“For RNA extraction, in order to limit the number of samples for sequencing, pellets were pooled as follows to obtain 7 samples per infection condition:”

R1.6

How was the RED score calculated for Figure 3C? As the RED score is dependent on a phylogenetic tree, which tree was used? Please add details to the methods section.

We apologize for this omission. The rooted phylogenetic tree, computed using OrthoFinder, is now presented as new Fig. S7. Additional details regarding this analysis have been included in the Methods section."

"Relative evolutionary divergence of megavirus chilensis genes

A rooted species tree was computed from *Imitervirales* genomes using OrthoFinder (v2.5.5)⁹³ with the "diamond_ultra_sens" option (Fig. S7). The following GenBank accessions were used: *Oceanusvirus kaneohense* (KY322437.1), *Tethysvirus hollandense* (KC662249.1), *Tethysvirus raunefjordenense* (KT820662.1), *Rheavirus sinuamexicani* (GU244497.1), *Fadolivirus algeromassiliense* (MT418680.1), *Theiavirus salishense* (MF782455.1), *Yasminevirus saudimassiliense* (UPSH00000000.1), *Cotonvirus japonicum* (AP024483.1), *Megavirus baoshanense* (MH046811.2), *Megavirus chilense* (JN258408.2), *Megavirus powaiense* (KU877344.1), *Mimivirus bradfordmassiliense* (HQ336222.2), *Mimivirus lagoaense* (KM982402.1), *Moumouvirus australiense* (MG807320.1), *Moumouvirus goulettemassiliense* (KC008572.1), *Moumouvirus moumou* (JX962719.1), *Tupanvirus altamarinense* (MF405918.2), *Tupanvirus salinum* (KY523104.2), *Kratosvirus quantuckense* (KJ645900.1). The obtained phylogenetic tree was consistent with the previously published phylogeny⁹⁴. We next computed relative evolutionary divergence (RED) scores⁵¹ using the "get_reds" function from the Castor R package⁹⁵. "

R1.7

L. 342 Supplementary Table 11 does not exist. Should likely read Supplementary Table 8B.

Indeed. Sorry for the confusion. This is now corrected.

Reviewer # 2

The study investigates the transcriptional interplay among four components in a complex host-virus system: a host (*Acanthamoeba castellanii*), a giant virus (*Megavirus chilensis*), a virophage, and a transposon. The aim is to decipher their transcriptional programs and interactions within the hyperparasitic system. In addition, proteomic data of the purified virions was also generated.

I think the manuscript is important as it explores the transcriptional complexity of hyperparasitism in a protist host, which will provide important baseline data and insights for future studies on hyperparasitism in diverse eukaryotes. I do not have any major concerns regarding the methods, analysis and interpretation. However, I believe some additional discussion is warranted regarding the transcriptional reprogramming of *Acanthamoeba* (see my comment below), along with some additional details in the method section.

Please see my comments below:

Thank you for your comments about this work.

R2.1

Regarding Method - Virus production and purification: Understandable that the authors have used the previously described method. However, a brief description of the approach would help the readers of the

manuscript who are mostly interested in the results and want some methodological context. I encourage adding some more details in these methods.

We understand the point. Thanks for this comment. We now give more details in the Methods section:

“Megavirus chilensis (GV), megavirus vitis and zamilon vitis were produced and purified as described ^{16,69}. Megavirus chilensis containing the transpoviron and zamilon vitis free of transpoviron were obtained from co-infection experiments carried out previously ¹⁶. Briefly, *A. castellanii* cells were co-infected with megavirus chilensis and zamilon vitis. After cell lysis, the culture was centrifuged at 10 000g for 25 min to pellet the giant virus and the supernatant containing zamilon vitis (Vp + Tpv) was purified as described ¹⁶. The pellet contained megavirus chilensis and zamilon vitis particles that stick to the giant virus. To separate them, the pellet was incubated in Tris buffer (40 mM pH 7.5) containing 250mM DTT at 55°C for 90 min. The resulting population of giant viruses was washed twice in Tris buffer and cloned ⁷⁰. Viral clones were screened by PCR to verify the presence of the transpoviron and the absence of virophage. One clone of megavirus chilensis containing the transpoviron (GV + Tpv) but free of virophage was recovered and amplified prior purification. To produce zamilon vitis free of transpoviron (Vp), purified particles of zamilon vitis were used to co-infect *A. castellanii* with the B-clade moutoumavirus maliensis that previously showed poor efficiency to replicate a C-clade transpoviron ¹⁶. The resulting population was cloned and clones were screened by PCR to isolate a sub-population devoid of transpoviron. The virophage particles devoid of transpoviron were separated from the giant virus by several steps of filtration/centrifugation and amplified with megavirus chilensis prior purification. All giant viruses and virophages preparations were controlled by negative staining observation after purification and the presence or absence of transpoviron was verified by PCR.

For mock-infections, viruses were inactivated by heat at 80°C for 7h.”

R2.2

Line 494: What was the MOI for? Is this MOI value specific to megavirus? In what concentration the virophage and transpoviron were added? How was the inoculum concentration of transpoviron measured?

The MOI reported here indeed refers to the giant virus. We only have a rough estimate of the virophages titer in our productions, based on the mean of previously counted preparations using NEMS by collaborators. Since virophages enter cells together with the giant virus, attached to its fibrils, the number of virophages that penetrate into the amoeba cell is unknown. It is likely limited to the number of particles able to adhere to the giant virus surface. We therefore carried out the infection using a large excess of virophage (about 100 virophages for 1 virus). The concentration of transpovirons added cannot be determined as the number of Tpv genomes loaded into each giant virus capsid is unknown. Moreover, the proportion of capsids (of both giant virus and virophage) filled with the transpoviron genome is also unknown.

We modified the text as follows:

“Cells were infected (12 flasks) or mock-infected (3 flasks) at MOI 50 for the giant virus and a large excess of virophage (about 100 for 1 virus) when applicable. The concentration of transpoviron is unknown. “

R2.3

Line 524,525: Was there a quality or confidence score cut-off that was used for Deeploc-2.0? Other parameters of the program should also be specified.

We used DeepLoc-2.0 with default parameters: high-quality model (slow) and long output format. For subcellular localization prediction, we applied recommended default cutoffs: cytoplasm (0.4761), nucleus (0.5014), extracellular (0.6173), cell membrane (0.5646), mitochondrion (0.6220), plastid (0.6395), endoplasmic reticulum (0.6090), lysosome/vacuole (0.5848), golgi apparatus (0.6494) and peroxisome (0.7364). This information has been added to the Methods section:

“Transmembrane domains and cellular localization were predicted for GV, Vp, and Tpv proteins with Phobius⁷⁴ (transmembrane domain > 0.8) and Deeploc-2.0⁴⁷ (Supplementary Table 5) with default parameters: high-quality model (slow) and long output format. For Deeploc-2.0 subcellular localization prediction, we applied recommended default cutoffs: cytoplasm (0.4761), nucleus (0.5014), extracellular (0.6173), cell membrane (0.5646), mitochondrion (0.6220), plastid (0.6395), endoplasmic reticulum (0.6090), lysosome/vacuole (0.5848), golgi apparatus (0.6494) and peroxisome (0.7364).”

R2.4

Regarding the transcriptional reprogramming of *Acanthamoeba* in response to giant virus infection - the authors should provide additional discussion to compare and contrast the response they saw in their study with the previous transcriptional studies - particularly members of Imitevirales groups and their hosts. For example, I believe one or more previous studies exist on transcription profile of *Acanthamoeba*-Mimivirus system:

<https://www.ncbi.nlm.nih.gov.insb.bib.cnrs.fr/pmc/articles/PMC2860168/>

<https://www.biorxiv.org/content/10.1101/2022.07.20.500700v2>

The Legendre et al. 2010 publication you mentioned, already cited in our manuscript (lines 195, 234, and 428), primarily focuses on viral transcriptional activity rather than host response. However, as suggested, we have revised our manuscript to include a more comprehensive comparison of the *Acanthamoeba castellanii* response to megavirus chilensis infection with the *Acanthamoeba polyphaga* infection by mimivirus (as described in the pre-print reference you referenced). A new paragraph detailing this comparison has been added:

“A similar transcriptional reprogramming of the host has been observed in *Acanthameoba polyphaga*, a related amoeba from the same genus, infected by mimivirus⁴⁰. Specifically, *A. polyphaga* genes involved in DNA replication and cytoskeletal remodeling are underexpressed during the mimivirus replication cycle. Similarly, *Acanthameoba* genes involved in transcription, translation regulation, and proteasome are activated in both *A. castellanii*/megavirus and *A. polyphaga*/mimivirus infections. However, several cellular genes associated with ribosome maturation, autophagy and protein folding are exclusively activated in the present study. This is likely due to an increased sequencing depth and temporal resolution of our host transcriptome analysis.”

We also mention both studies from the giant virus perspective in this new paragraph:

“Overall, megavirus chilensis exhibits expression profiles of key functions that are similar to mimivirus during the replication cycle, whether infecting *A. castellanii*²⁵ or *A. polyphaga*⁴⁰. After 3 hours post infection, viral genes involved in DNA replication and transcription are highly expressed, and at the end of the infection cycle, genes associated with sugar metabolism, collagen, and capsid production are expressed in both viruses.”

R2.5

Line 257-259: I believe this has also been suggested in several more studies. See:

[https://www-cell-com.insb.bib.cnrs.fr/trends/genetics/abstract/S0168-9525\(06\)00367-2?large_figure=true](https://www-cell-com.insb.bib.cnrs.fr/trends/genetics/abstract/S0168-9525(06)00367-2?large_figure=true)

<https://www-sciencedirect-com.insb.bib.cnrs.fr/science/article/pii/S0042682214002992>

Thank you for pointing out these references. We have added them to the manuscript, as well as others, including one suggested by reviewer #3 (see R3.4).

R2.6

Line 267-270: As per the data, it seems like the vast majority of host transcripts were suppressed during the late stage of infection and viral RNA dominated the virocell, not only the ones that have homologs in the virus. Since the host transcript had a global suppression, I am not sure if a causal connection between the host and virus homologs can be made - the suggestion here is that the viral homologs possibly ‘replaced’ the functions of the host homolog - but we can not rule out the possibility that the viral homologs had a function that is distinct from the host counterparts. Further discussion could be added to clarify this.

We appreciate this insightful comment. You are right, a more cautious interpretation to virus/host homologs expression is necessary. We have revised the main text to highlight the potential functional differences between these homologs.

“As previously described, the majority of host transcripts exhibit decreased expression levels during the late stages of infection. This includes genes with viral homologs, such as those involved in transcription, which are differentially expressed in both, the host (Fig. 2) and the GV (Fig. 3A). Focusing on shared transcription-related genes, we found that their expression levels usually overlap towards T2-T3, but while host gene expression drastically drops right after, the expression of virally-encoded homologs is generally maintained until the end of the infectious cycle (Fig. S8). Assuming viral homologs preserve cellular functions, like the poxviruses-encoded DNA-dependent RNA polymerase⁴⁴, transcriptional capacity of the virocell might be maintained by GV compensation. Nevertheless, there are numerous examples in giant viruses of virally-encoded homologs that evolved distinct functions from their cellular counterparts^{33,56}. Further studies on the megavirus-encoded transcriptional machinery components will thus be required to explore their role within the virocell during infection.”

R2.7

Line 301: ‘peculiar’ or ‘particular’?

“Particular” is indeed the proper term, thank you for the correction.

R2.8

General comment: Please ensure scientific names are italicized throughout the manuscript. For example, “*Acanthamoeba*”.

We have italicized the species names as requested. However, for virus names, we have followed the ICTV rules (<https://ictv.global/faq/names>), that distinguishes virus names (not italicized) from viral species names (italicized).

R2.9

(Remarks on code availability):

I believe the script used to analyze the data is made available in figshare, although I didn't find a readme with further instructions on how to use these scripts.

We have included a README file containing essential information for running the analyses.

Reviewer # 3

This manuscript by Bessenay et al investigated the outcome of an evolutionary interplay of amoeba, *Acanthamoeba castellanii*, its giant virus, megavirus chilensis with its virophage, Zamilon vitis, and a transpoviron (a genetic parasite) known as megavirus vitis transpoviron. Some giant viruses, such as the megavirus vitis used in this study, carry linear extrachromosomal elements of about 7 kb coding for eight genes. They are known as transpovirons. Virophage and transpoviron are dependent on giant viral transcription machinery.

The authors have carried out extensive transcriptional analysis (RNA sequencing) at various time points during the infection to assess the outcome of the interaction of these four biological entities during a single infection cycle (up to 12 h, in triplicates). They show that, as expected, giant viruses significantly modify the host transcription profile. Interestingly, while coinfection with virophage resulted in transient changes in the giant viral transcriptome with no significant effect on either the fitness of the giant virus (giant virus production from viral factories) or the protein composition of the assembled giant viral particles, it led to the “overexpression” of the transpoviron genes. Just 30 min post-infection, a significant change was observed in the expression levels of host transcriptome with enhanced expression of Rho family GTPases, which could be involved in the host cytoskeleton remodeling. They have also observed a lowered gene expression in the host cell encystment. It appears that the presence of virophage enhances the expression of Tpv genes.

The experimental strategy is comprehensive and gives some insights into the changes in the transcriptional profile in a four-way hyperparasitic system and when one or more players are absent.

R3.1

While most of these insights are, at best, confirmatory to what has been already known, the study lacks mechanistic or novel insights. Some hypothesis and follow-up experiments (long-term or otherwise) would have helped in gaining those insights. While the evolutionary outcome of a four-way host-nested parasite is complex and probably context-dependent, any generalizations (some hypotheses) would have added more value to this study. For example, it is well-known that host-parasite interactions are major drivers of

diversity. Any insights on how or whether this four-level parasitism could lead to the diversification of giant viruses, or some thoughts on the loss of virulence and its consequence (if my interpretation of the data is correct) of Tpv would have greatly enhanced the manuscript.

Our study aimed to elucidate the complex transcriptional regulations involved within the four-component hyperparasitic system, which to our knowledge has not been studied yet. While the evolutionary implications of the interactions between the partners is of major interest and related to this work, our experimental design was not optimized to address them. A dedicated, long-term evolutionary experiment would indeed be necessary to fully explore these consequences, which is beyond the scope of this study. However, to get insights on how the transpoviron and virophage could contribute to GV diversity, we have performed long read nanopore DNA sequencing of megavirus vitis, zamion vitis and megavirus vitis transpoviron. This added data shows several occurrences of chimeric reads covering the megavirus vitis GV genome and the zamion vitis or megavirus vitis transpoviron genomes. This likely corresponds to genomic integration events of Vp and Tpv into the GV genome, as observed for mimivirus (Desnues et al. 2012, <https://doi.org/10.1073/pnas.1208835109>). This is now mentioned in the results section and in a supplementary figure (new Fig. S9):

“To investigate potential Tpv and Vp integration into the GV genome, we sequenced using Nanopore long reads the genomic DNA of megavirus vitis, a closely related GV strain (97.9% average nucleotide identity with megavirus chilensis) from which zamion vitis and megavirus vitis transpoviron were isolated ¹⁶. We identified 12 megavirus vitis chimeric reads aligning to Tpv (Fig. S9A), and 2 to Vp (Fig. S9B), suggesting potential Tpv and Vp insertions within the megavirus genome. These insertions appear uniformly distributed throughout the genome (K-S test against uniform distribution P value = 0.621), similar to observations in mimivirus ²³. These findings suggest potential GV diversification resulting from Tpv and Vp insertions. However, the low number of chimeric reads and their occurrence within essential genes (e.g., major capsid protein 3, mRNA capping enzyme, Fig. S9A) indicate rare events probably often leading to evolutionary dead ends.”

We also provide a more detailed discussion on this topic:

“While this study focused on a specific system, GV-related hyperparasitic systems encompass a diverse array of virophages and cellular hosts ^{10,11,13}, exhibiting varying degrees of positive, negative or neutral interactions. Other giant virus/virophage pairs, such as sputnik/mimivirus and mavirus/crov, demonstrate more pronounced parasitic relationships. It is tempting to speculate that the observed zamion/megavirus transient hyperparasitic interaction might represent an evolutionary transition from strict parasitism towards commensalism ⁶⁵. This hypothesis is supported by experimental coevolution studies on mavirus and crov, showing increased virophage replication with decreased giant virus exploitation ⁶⁶. Similarly, the transpoviron commensal status confirmed in our study could reflect an adaptation from a more parasitic role. To fully understand the impact of the transpoviron and potential loss of virulence, analysis within a more parasitic system like mimivirus/sputnik would be necessary ¹⁰. Regardless, as previously observed, both transpovirons and virophages ²³, can integrate into GV genome (Fig. S9), directly contributing to giant virus genomic diversity.

While the quadripartite system we analyzed in this work is complex, it represents only a portion of a larger ecosystem. *Acanthamoeba* hosts various symbionts and pathogens ⁶⁷ that can interact with infecting giant viruses ⁶⁸. Not to mention that *Acanthamoeba* is by itself an opportunistic pathogen of humans and animals. We are thus faced with a fascinating theater of interlocking parasites.”

R3.2

Minor comments

The Results and Discussion section can be just "Results" and the "Conclusion" is too long, it can be the "Discussion" section as the findings from the study have been captured with good context.

We have modified the section titles as suggested.

R3.3

In the Abstract, the authors have mentioned nested parasitism, but in the introduction, they talk about hyperparasitism. These terminologies can be a bit confusing, especially to the uninitiated in the host-parasite coevolution. I think there is a subtle difference in the concepts and the authors may clarify them in the introduction.

From previous studies on giant virus/virophage, the terms "nested parasitism" and "hyperparasitism" are often used interchangeably (e.g. Duponchel & Fischer, 2019, <https://doi.org/10.1371/journal.ppat.1007592>). However, we acknowledge that "nested parasitism" might not be the most accurate descriptor. Consequently, we removed this term from the manuscript and consistently employed "hyperparasitism" throughout.

R3.4

Lines 255-259: The presence of the so-called "core genes" towards the middle of the genome has also been shown in all NCLDVs by a 2016 paper (PubMed ID 29308275) that can also be cited to support the observation, in addition to reference 51.

We thank you for the suggested reference, which we have incorporated alongside other references from Reviewer #2 on the same topic (see R2.5).

R3.5

The sentence (338-341), "As for Tpv coinfection, Vp effect on the host transcriptome is negligible, with only six genes differentially expressed, of which four are weakly expressed (average expression < 5 TPM, Supplementary Table 5A and Supplementary Table 8A)" can go towards the end of the same paragraph for better readability.

In this paragraph, we detail the impact of Vp on the virocell transcriptome by first outlining its negligible effect on the host cell and then its strong effect on the GV. The subsequent paragraph provides a more detailed analysis of the timing of the Vp's effect on GV gene expression. Therefore, we believe that moving the sentence (338-341) would interfere with the paragraphs' continuity. However, we understand your concerns about the overall clarity and revised the paragraphs. It now reads as follows:

"To determine the Vp's impact on the virocell transcriptome, we next compared our transcriptomic data in the presence and absence of Vp (i.e., C+GV+Vp vs C+GV, Fig. 1A). Our analysis revealed a negligible impact of Vp on the host transcriptome, with only 6 cellular genes differentially expressed, 4 of which were weakly expressed (average expression < 5 TPM, Supplementary Table 5A and Supplementary Table 8A).

In contrast, Vp strongly disrupted GV gene expression, significantly altering the expression of 23% (263/1134) of its genes (Supplementary Table 8B). This substantial effect could be attributed to a bias arising from the introduction of a new partner with a finite pool of sequenced reads. As a control, we performed the same analysis excluding the Vp genome sequence from the mapping, i.e. only C and GV

reference sequences were included. After confirming sufficient read coverage (Fig. S3), we still found that 22% (254/1134) of GV genes were differentially expressed. Thus, the observed differential expression of GV genes is indeed a result of its interaction with Vp, and not due to a bias in the proportions of mappable reads.

The effect of Vp on GV gene expression is mainly negative, as most differentially expressed GV genes (238/263) are underexpressed in its presence. This mainly occurs at T4 (Fig. 5B), at the same time as peak expression for most Vp genes (Fig. 5A and Supplementary Table 9). Competition for transcription machinery might thus occur between the two viruses (GV and Vp) at this timepoint. This is supported by the fact that Vp genes are globally more efficiently expressed than GV genes (Fig. S10). ”

R3.6

Lines 398-405: The delay in the expression of some Tpv genes in the presence of Vp is a good insight from the study. It is not clear whether there is also a concomitant delay in the replication of the transpoviron in the presence of Vp. Also, in the absence of Vp, how many copies of transpovirons are made, and how the copy number (a determinant of its fitness) is affected in the presence of Vp.

This is an interesting question. To investigate potential differences in Tpv DNA replication timing and efficiency with and without Vp, precise quantification of Tpv genomes within Vp and GV particles under both conditions would be necessary. Unfortunately, as explained in our response to reviewer # 2 (R2.2), the exact number of Tpv genomes per capsid and the proportion of filled capsids remain unknown, preventing us from addressing this question.

R3.7

The selective packaging of transpoviron (again if my understanding is correct) into Vp rather than GV is interesting indeed, but it has to be experimentally confirmed. Can there be a scenario where transpoviron is present in both GV as well as Vp? If yes, can this also be considered?

The transpoviron can indeed be package within both, GV and Vp particles. Zamilon vitis and the transpoviron were originally found associated to the megavirus vitis GV strain (Jeudy et al. 2020, <https://doi.org/10.1038/s41396-019-0565-y>). In that case, the Tpv present in both particles was inseparable from GV particles. To circumvent this, we employed the megavirus chilensis GV, a very closely-related strain to megavirus vitis, that is free of Tpv. This allowed to have a GV without Tpv as control. Therefore, in the condition where all partners are present, the GV is exclusively Tpv-free. The presence or absence of Tpv in the different preparations used in this study have been verified by PCR. The experimental procedure employed is now detailed in the Methods section (see R2.1).

REVIEWERS' COMMENTS

Reviewer #1 (Remarks to the Author):

All of my previous comments have been addressed adequately in this revised manuscript version.

Reviewer #2 (Remarks to the Author):

Thank you for addressing my comments. I have no further concern.

Reviewer #2 (Remarks on code availability):

The scripts and necessary files are available, which should facilitate reproducing aspects of the analysis.

Reviewer #3 (Remarks to the Author):

The authors have addressed all the concerns and thoroughly modified the manuscript. I have no further queries.

Reviewer #3 (Remarks on code availability):

Not applicable